

# Towards using state-of-the-art climate models to help constrain estimates of unprecedented UK storm surges

Tom Howard[Met Office Hadley Centre] and Simon David Paul Williams[National Oceanography Centre]

[1]Met Office, FitzRoy Road, Exeter EX1 3PB, UK
[2]National Oceanography Centre, Joseph Proudman Building, 6 Brownlow Street, Liverpool L3 5DA, UK

**Correspondence:** Tom Howard (tom.howard@metoffice.gov.uk)

**Abstract.**

Our ability to quantify the likelihood of present-day extreme sea level (ESL) events is limited by the length of tide gauge records around the UK, and this results in substantial uncertainties in return level curves at many sites. In this work, we explore the potential for a state-of-the-art climate model, HadGEM3-GC3, to help refine our understanding of present-day coastal flood

risk associated with extreme storm surges, which are the dominant driver of ESL events for the UK and wider European shelf seas.

We use a 483-year present-day control simulation from HadGEM3-GC3-MM (1/4 degree ocean, approx 60 km atmosphere in mid-latitudes) to drive a northwest European shelf seas model and generate a new dataset of simulated UK storm surges. The variable analysed is the skew surge (the difference between the high water level and the predicted astronomical high tide),

which is widely used in analysis of storm surge events. The modelling system can simulate skew surge events comparable to the catastrophic 1953 North Sea storm surge, which resulted in widespread flooding, evacuation of 32 thousand people and hundreds of fatalities across the UK alone, along with many hundreds more in mainland Europe. Our model simulations show good agreement with an independent re-analysis of the 1953 surge event and suggest that a skew surge event of this magnitude has an expected frequency of about 1 in 500 years at the mouth of the river Thames. For that site, we also revisit the assumption

of skew surge/tide independence. Our model results suggest that at that site for the most extreme surges, tide/surge interaction significantly attenuates extreme skew surges on a spring tide compared to a neap tide.

Around the UK coastline, the extreme tail shape parameters diagnosed from our simulation correlate very well (Pearson's r greater than 0.85), in terms of spatial variability, with those used in the UK government's current guidance (which are diagnosed from tide-gauge observations), but ours can be diagnosed without the use of a subjective prior.

Despite the strong correlation, our diagnosed shape parameters are biased low relative to the current guidance. This bias is also seen when we replace HadGEM3-GC3-MM with a reanalysis, so we conclude that the bias is likely associated with limitations in the shelf sea model used here.

Overall, the work suggests that climate model simulations may prove useful as an additional line of evidence to inform assessments of present-day coastal flood risk.


## 1   Introduction

Around £150 billion of assets and 4 million people in the UK are at risk from coastal flooding (Haigh et al., 2017), and estimated damages to the UK from coastal flooding are of the order of £500 million per year (Edwards, 2017). It is neither technically feasible nor economically affordable to prevent all such flooding, so policymakers use a risk-based approach. Typically, coastal flood protection is mandated based on an extreme high water "return level" with an estimated average

recurrence interval, which is the expected average time between exceedances of that level (conceived as averaged over a period including many such exceedances). The average recurrence interval is sometimes called a "return period". We use these names interchangeably here, although some authors use a different definition of the return period. Typically, assets with high value and/or high vulnerability will have a mandate for protection against a return period of a thousand or even ten thousand years. Typical tide gauge records cover much shorter periods (of the order of 30 to 150 years). To address this, the traditional

approach is to fit a statistical extreme value model in order to extrapolate from the observations. Many different statistical approaches have been used (Haigh et al., 2010; Batstone et al., 2013); see §3.3. However, even using the current best practice, the inevitable extrapolation involved means that the uncertainties in the magnitude of very rare (perhaps unprecedented) events may be very large. For example, the size of the 90 % confidence interval on the 10,000-year return level at Sheerness is around 1.6 metres (Environment Agency, 2018). For comparison, the 90 % confidence interval on model projections of regional mean

sea level rise to 2100 relative to the 1981-2000 average under representative concentration pathway RCP8.5 for the same location is around 0.62 metres (Palmer et al., 2018). Coles (2001) discusses some of the advantages and disadvantages of the statistical modelling approach; he says: "*Caution is required in the interpretation of return level inferences especially for return levels corresponding to long return periods... estimates and their measures of precision are based on an assumption that the model is correct.*" The statistical models which are fitted to the observational data in order to infer the levels of unprecedented

extremes are supported by mathematical arguments which may require assumptions such as the assumption that the events are effectively random, and statistically independent of each other. We know that the real-world events are deterministic, and may in reality be auto-correlated over a range of timescales. Although some account can be taken of this within the statistical framework, for example by the use of an extremal index (Tawn, 1992; Batstone et al., 2013), a physically-based numerical model has the potential to directly address these issues by simulating them. Coles (2001) goes on to say: " *Though the* [extreme

value statistical] *model is supported by mathematical argument, its use in extrapolation is based on unverifiable assumptions, and measures of uncertainty on return levels should properly be regarded as lower bounds that could be much greater if uncertainty due to model correctness were taken into account.*"



An alternative approach is to exploit a physically-based numerical model of the coastal shelf waters. Such models typically parameterize the surface stress associated with winds and pressure from an atmospheric forecast model, and are routinely used to make short-range (e.g. less than 48-hour) forecasts of storm surges whenever a potentially hazardous atmospheric storm is identified in the atmospheric forecast. Bernier and Thompson (2006) found that when the atmospheric forecast model is replaced by an atmospheric hindcast, realistic extreme storm surge events were simulated in the northwest Atlantic.

Another approach is to make plausible modifications to the strength, track, or speed of selected observed atmospheric events, and use the resulting simulated atmospheric forcing to drive the coastal shelf model (Brown et al., 2010). Very recently Horsburgh et al. (2021) used this approach. They selected the storm of 5th December 2013 and made manual adjustments to the quasi-geostrophic potential vorticity field, inverting it to get dynamically self-consistent fields of sea-level pressure and wind. They showed that this approach can produce synthetic surges which are substantially larger than any in the observational record, for sites on the UK east coast. However this approach does not offer a way of quantifying the probability of the synthesized events.

Yet another approach, adopted here and discussed further in §3.4, is to simulate extreme atmospheric events using a physically based numerical climate model, which in turn is used to drive the coastal shelf model.

An obvious advantage of this approach is that the model is based on verifiable real-world physics. Many climate model simulations extend over periods longer than the tide-gauge record. In particular, in order to evaluate model performance, modellers use control simulations (with greenhouse gas forcing fixed at either pre-industrial or present-day levels) which may extend over many hundreds or even thousands of years. Ensemble simulations provide another potential source of data effectively covering a much longer period than the observations. Using the data from such simulations provides a further line of evidence in the effort to predict the magnitude and frequency of unprecedented events. This method was applied to seasonal rainfall totals in the UK (Thompson et al., 2017), and Grabemann et al. (2020) applied the method to extreme storm surges for locations in the German Bight, successfully identifying a number of simulated water levels exceeding those in the observational record since 1906.

This article reports a preliminary investigation into the value of using this approach to help form return level curves of storm surge around the UK coast with a view to providing improved likelihood information on the most extreme coastal water levels.

**Climate change**

Mean sea level is increasing, and will continue to increase, both at UK national scale (Palmer et al., 2018, 2020) and at global scale (Pörtner et al., 2019), and this will exacerbate future coastal flood risk. However, for many locations around the UK (exemplified by Sheerness as described above) the uncertainty in the projections of future mean sea-level rise is not as large as the uncertainty associated with, say, the 1000-year return level of storm surge and it is the effort to reduce this larger uncertainty that we are concerned with here: we are trying to "focus the snapshot" of conditions in the current climate. Thus we do not explicitly address mean sea level change in this work, but rather we note that the effects of mean sea level change and its uncertainty will need to be considered in addition to the present-day hazard which we discuss here, for example through the use of a sea-level rise allowance (Howard and Palmer, 2020). Also, we do not consider the effects of long-term change in the





mean strength or location of the North Atlantic storm track (Shaw et al., 2016; Shepherd, 2014). Many studies (e.g. Palmer et al., 2018; Lowe et al., 2009; Sterl et al., 2009; Howard et al., 2019) have suggested that the change in local mean sea level will be the main contributor to the changes in the sea level extremes, as it has been in the past (Menéndez and Woodworth, 2010), with the change in the storm track making a smaller secondary contribution. We do not consider this secondary contribution here.

## 2 Nomenclature and Notation

For ease of reference, some terms which arise throughout this article are given in table 1.

| Acronym or Symbol | Description |
|---:|---|
| CS3 | Continental Shelf 3: our North-West European storm surge model. See §3.1. |
| HadGEM | Hadley Centre Global Environment Model. See §3.4. |
| CFB2018 | Coastal Flood Boundary Conditions for the UK: update 2018 (Environment Agency, 2018). |
| CMIP5 | Climate Model Intercomparison Project, Phase 5 (Taylor et al., 2012). |
| CMIP6 | Climate Model Intercomparison Project, Phase 6 (Eyring et al., 2016). |
| SWL | Still Water Level. Still water level includes the astronomical tides and surge but does not include the short-period oscillations due to waves. See §3.2. |
| GEV, GEVD | Generalised Extreme Value (Distribution) (Coles, 2001). See §3.3. Under appropriate conditions, annual maxima are expected to follow a GEVD. |
| GPD | Generalised Pareto Distribution (Coles, 2001). See §3.3. Under appropriate conditions, all extreme values over a high threshold are expected to follow a GPD. |
| POT | Peaks Over Threshold. A POT model uses all values over a high threshold (see GPD). |
| MLE | Maximum Likelihood Estimator (Coles, 2001). See §3.3. |
| PMLE; GMLE | Penalised Maximum Likelihood Estimator; Generalised Maximum Likelihood Estimator. Used synonymously here. See §3.3. |
| prior; penalty; constraint | These all refer to PMLE. See §3.3. |
| $\mu$ | GEV location parameter. See §3.3. |
| $\sigma$ | GEV scale parameter. See §3.3. |
| $\xi$ | Shape parameter. See §3.3. |
| $\tilde{\sigma}$ | GPD scale parameter. See §3.3. |
| $R$ | Return Period in years. See §3.3. |
| $L$ | Natural logarithm of Return Period. See §3.3. |
| $y$ | Return Level. See §3.3. |

**Table 1.** Acronym and Symbols



## 3 Models, Methods and Data Sources

### 3.1 The CS3 coastal shelf model

Our barotropic coastal shelf model, CS3 (Continental Shelf 3, Horsburgh et al., 2008; Flather, 2000, 1994) is very similar to the CS3X (Continental Shelf 3 Extended) model which until very recently was used in the UK operational storm surge forecast/warning system. The domain and grid of CS3 are shown in the appendix in Fig. A1(a). The model produces a numerical solution of the discretized nonlinear shallow water equations with friction. The model is barotropic in the sense of solving the depth-averaged equations (i.e. it is a two-dimensional model). The horizontal resolution is approximately 1/9 degree latitude by 1/6 degree longitude (approximately 12 km). The model has been shown to perform particularly well during extreme storm surges in the southern North Sea (Horsburgh et al., 2008), forecasting surge in the Thames estuary to within 10 cm when driven by re-analysed meteorology. CS3 is "one of the most validated operational storm surge forecasting models in the world" (Horsburgh et al., 2021). Further details of storm surge model evaluation can be found in Furner et al. (2016); O'Neill et al. (2016); Palmer et al. (2018); Flather (2000). Typical RMS errors when forced with numerical weather prediction model atmospheric data are of the order of 10 cm.

### 3.2 Coastal Flood Boundary Conditions for the UK: update 2018

Coastal Flood Boundary Conditions for the UK: update 2018 (Environment Agency, 2018) (henceforth CFB2018) contains the latest UK government best estimates and uncertainty estimates for the distribution of extreme still water level (SWL) under present-day mean sea level. SWL can be thought of as the water level averaged over about five minutes to remove the short-period oscillations due to surface waves. It includes the astronomical tide and storm surge, and is the level that is reported at the tide gauges. The CFB2018 approach is based on data from tide-gauge observations, without reference to model simulations. Discussion of their approach is included below.

### 3.3 Statistical Modelling of Extreme Values

To identify, for example, the 1000-year return level based solely on tide-gauge observations, some philosophy for making out-of-sample estimates is required. The usual approach is to exploit the most extreme observations, and theories concerning their behaviour, under some restrictive assumptions.

**Annual Maxima**

One popular and simple approach is fitting a Generalised Extreme Value (GEV) distribution to the annual maxima. The GEV distribution (GEVD) arises as the limiting case for block maxima as the block size tends to infinity. In the case of annual maxima, "block" means one year. The GEVD is characterised by three parameters. For readers unfamiliar with the GEVD, it may be helpful to picture the effect of these parameters in terms of a return-level curve, such as the ones shown in Fig. 1. The location parameter, $\mu$, is comparable to an intercept. An increase in $\mu$ slides the whole curve up the Y-axis. $\mu$ is the Y-value



(return level) evaluated at the one-year return period. The GEV scale parameter, $\sigma$, is the gradient of the curve, evaluated at the
one-year return period:

$$\sigma = \left.\frac{dy}{dL}\right|_{L=0} \tag{1}$$

where $L = \log(\text{return period})$ and $y$ is the return level. The shape parameter, $\xi$, determines the curvature. Negative $\xi$ corresponds to a curve which flattens out at high return periods, approaching an upper bound as the return period tends to infinity. With positive $\xi$ the curve has no upper bound, but has a lower bound as the return level decreases. When $\xi = 0$ the curve is a straight line and has neither lower nor upper bound. This follows the convention of Coles (2001) for the shape parameter. However, not all sources follow this convention. In CFB2018, "shape parameter" refers to the negative of our $\xi$. In the wider literature the "shape parameter" may refer to the negative or the reciprocal of our $\xi$. To make our shape parameter notation unambiguous: if $Y$ is a random variable with GEV distribution, our shape parameter $\xi$ is defined such that the distribution of $Y$ is given by

$$P(Y < y) = \exp\left\{ -\left[1 + \xi\left(\frac{y - \mu)}{\sigma}\right)\right]^{-1/\xi} \right\} \tag{2}$$

This can be more simply expressed as the corresponding return level curve, which is

$$\frac{y - \mu}{\sigma} = \frac{R^\xi - 1}{\xi} \tag{3}$$

where the average recurrence interval (or "return period") is $R$ and the corresponding return level is $y$. The connection between equations 2 and 3 is seen by regarding exceedances of the $R$-year return level $y$ as Poisson-distributed random occurrences, occurring at an average rate

$$\lambda = 1/R \tag{4}$$

The probability of no such occurrences in a given year is then given by standard Poisson statistics:

$$P(\text{no occurrences}) = P(Y < y) = \exp(-\lambda) \tag{5}$$

Combining 3, 4 and 5 gives equation 2. The particular case $\xi = 0$ is obtained by taking the limit as $\xi \to 0$.

**Peaks over Threshold**

The most extreme storm surges in the UK are caused by the storminess of the winter atmosphere, so the annual maximum event is always expected to occur in winter. Thus, an advantage of the annual-maxima approach described above is that the annual maxima are typically very well separated from each other and thus can be considered independent, particularly if the nominal year change is taken to be in the summer. A disadvantage of the approach is that it uses only the annual maxima. On the other hand, the peaks-over-threshold (POT) approach uses all of the data exceeding a chosen threshold. This formed part of the approach taken by CFB2018 (Environment Agency, 2018). An advantage of this approach is that, if a low-enough


threshold is used, it has the potential to exploit more of the available data (i.e. an average of more than one extreme event per year), whilst including only extreme events. Such exploitation of more data usually reduces the uncertainties in inferred statistics (e.g. the out-of-sample estimates). This is particularly desirable when short observational records limit the available extremes. However, if the threshold is too low, some of the data included can no longer be considered "extreme" and may bias the result. This is the well-recognised bias-variance trade-off. Another disadvantage is that including more than one event from a winter may compromise the independence of the events. (Skew surge can be evaluated for every high tide, and a weather system can generate a substantial skew surge on successive high tides.) Dependence is accommodated by CFB2018 using an extremal index (Tawn, 1992; Batstone et al., 2013). For a detailed comparison of the annual-maxima and POT approaches see Arns et al. (2013).

The usual POT approach is to fit a Generalised Pareto Distribution (GPD) to the peaks. The GPD has two parameters. The shape parameter $\xi$ is shared with the GEVD. The GPD scale parameter, $\widetilde{\sigma}$, is the gradient of the plot of return level against log of return period at the return period of the chosen threshold, $u$,

$$\widetilde{\sigma} = \frac{dy}{dL}\Big|_{y=u} \tag{6}$$

This is a property of both the extreme value distribution and the chosen threshold. The GEV scale parameter, $\sigma$, on the other hand, is a property of the extreme value distribution only and is thus a more fundamental parameter for making comparisons: it can be used in a like-for-like comparison of the results of different thresholds, or for comparison of GEV and GPD results. The two different scale parameters are related by $\sigma = \widetilde{\sigma}\lambda_u^{\xi}$, where $\lambda_u$ is the expected number of exceedances of $u$ per year.

Though not formally a parameter of the GPD, a threshold must be chosen. CFB2018 tested 14 different thresholds and, finding no clear support for dismissal of any, elected to evaluate statistics based on each threshold and identify the median as the best estimate.

**Maximum Likelihood Estimation**

As a model-fitting approach, CFB2018 adopted maximum likelihood estimation (MLE, Coles, 2001) and so do we.

**Penalised Maximum Likelihood Estimation/Generalised Maximum Likelihood Estimation**

A recognised problem of short records such as the relatively short tide-gauge record at some sites is the diagnosis of "noisy" and implausible shape parameters by MLE (see appendix C). We also show in appendix C that the uncertainty in estimating unprecedented events from observational records using MLE is dominated by uncertainty in the shape parameter. One fix for this is to put a subjectively-chosen prior (or "penalty function") on the shape parameter (Coles and Dixon, 1999; Martins and Stedinger, 2000). This method was used by CFB2018. It is variously known as Generalised Maximum Likelihood Estimation or Penalised Maximum Likelihood Estimation (PMLE). We also refer to the penalty function as a constraint. We show below that the need for a penalty function is obviated in the case of our simulation, due to the long record lengths. We argue that this removes some of the subjectivity.



**Skew Surge Joint Probability Method**

The large return-level uncertainties for long return-period events are mitigated by the use of the skew surge joint probability
method, the current state-of-the-art approach. Extreme SWLs are composed of a high astronomical tide and a meteorological
surge. The metric of choice for the meteorological component is the skew surge (de Vries et al., 1995): the difference between
the (deterministic, predictable) astronomical high tide and the actual high water level (which typically arrives at a slightly
different time). See Palmer et al. (2018) for a schematic diagram illustrating the definition of skew surge (their figure A1.3.4).
Under the assumption of tide-skew surge independence, which has substantial observational support (Williams et al., 2016),
the level of the high tide is assumed to have no effect on the magnitude of the skew surge and thus any skew surge can combine
with any high tide. This suggests a method (exploited by CFB2018) whereby the observed surges are decomposed into tide
and skew surge to give a skew surge distribution, which can be convolved with the full, known distribution of high tides to
form the full distribution of high water levels. The extreme value modelling is only involved in establishing the high tail (i.e.
the outside-sample part) of the skew surge distribution. The implication of this convolution is that although a very rare high
water level might be a combination of an equally rare skew surge and an ordinary tide, it could also be formed by a very rare
high tide (the distribution of which is well known) and an ordinary skew surge. A consequence is that uncertainties in very rare
high water levels map to the uncertainties in less-rare skew surges, and these uncertainties are smaller than the uncertainties in
very rare skew surges. In other words, for a given return period, the high-water-level uncertainty is smaller than the skew-surge
uncertainty. This is good, because it is the high water level that we are concerned about from a coastal flooding point of view.
Having said all that, we do not have cause to use the skew surge joint probability method in this work; we only mention it due
to its relevance to the CFB2018 estimates. We revisit the assumption of tide-skew surge independence in §5.2.

### 3.4 A free-running climate model as a driver of synthetic storm surges

The atmospheric jet over the north Atlantic, which is associated with the extratropical cyclones which drive surges on the
UK coast, has complex variability with a trimodal latitudinal behaviour (Woollings et al., 2010). A lot of effort in the climate
modelling community is directed to understanding and improving the quality of models' simulation of this behaviour, owing to
its importance in projections of climate change in the mid-latitudes (Shaw et al., 2016; Shepherd, 2014). Ongoing improvements
in the representation of the North Atlantic storm track in global climate models are discussed by Roberts et al. (2018); Priestley
et al. (2020).

Williams et al. (2015) show improvements in the representation of storm tracks in the CMIP6 (Eyring et al., 2016) gen-
eration Hadley Centre models relative to HadGEM2-AO (the Hadley Centre model which contributed to CMIP5), with both
HadGEM3-GC2 and HadGEM3-GC3 simulating the winter latitudinal variability well. Both models employ the ENDGame
revision to the dynamical core, which reduces the numerical damping associated with the semi-implicit advection scheme
and has been shown to increase synoptic variability (Williams et al., 2015). This suggests that a surge simulation driven by
HadGEM3-GC3 surface wind and pressure might yield realistic storm surges for the UK. We exploited a 483-year control
simulation of HadGEM3-GC3-MM (Williams et al., 2018, and references therein). In this simulation, greenhouse gas concen-





trations are fixed at pre-industrial levels. Its atmospheric horizontal resolution is N216 (approximately 60 km in mid-latitudes), and ocean horizontal resolution approximately 1/4 degree. The atmospheric component (Walters et al., 2019) of this model exhibits a very good representation of the storm track (as measured against the ERA Interim (Dee et al., 2011) reanalysis) when forced with present-day sea-surface temperatures (pers. comm: Julia Lockwood, by email).

One argument that might be made against this approach is that the spatial resolution of the global climate model may be inadequate to resolve all of the physical processes that might be important in generating extreme events, particularly small-scale extremes. For example, contemporary global climate models do not have adequate resolution to synthesize a small convective event such as a thunderstorm. However, three factors argue against this being a problem in the case of UK storm surge modelling:

– Storm surge in the UK is usually driven by atmospheric baroclinic instability, which is a large-scale process, much larger than the scale of a single thunderstorm, and well-captured by atmospheric models.

– Storm surge effectively integrates the driving atmospheric wind and pressure over a large area and time (Sterl et al., 2009), so that shortcomings in the simulation of small temporal- or spatial-scale phenomena are relatively less important than they would be in the simulation of, for example, localised short-duration extreme rainfall events.

– Storm surge generation occurs over the sea. It has been well-recognised for some time that the orographic drag schemes used in atmospheric modelling improve the column-average windspeed at the expense of realistic surface windspeeds over high ground (e.g. Howard and Clark, 2007). On account of this we might expect to find issues with simulation of the surface wind climatology over land, particularly over high ground. Whilst this issue might also affect the ocean points nearest to the coast, it has little effect over the open sea, where most of the surge is generated. de Winter et al.
(2013) evaluated 12 global climate models in terms of their simulation of wind over the North Sea, including two models from the HadGEM family. Their results show that these two models (along with two models from the GFDL-ESM family) exhibit a particularly realistic distribution of extreme winds (evaluated against a reanalysis), being well within the uncertainties of the reanalysis.

The UK Climate Projections 2018 Marine Report (Palmer et al., 2018) provides extensive evidence of the realism of storm
surges simulated by CS3 when driven by climate model winds and pressure.

## 4    Results and Discussion

Example empirical return level plots of skew surge, comparing model and observational (tide-gauge) data at two sites are shown in Fig. 1. We take the model grid cell closest to the location of the real-world tide-gauge to represent that tide-gauge. The two sites chosen for this illustration are Sheerness, at the mouth of the river Thames in south-east England, a site of great
economic and societal importance, and Workington, a coastal site in the north west of England which is typically affected by different storms (Haigh et al., 2016).





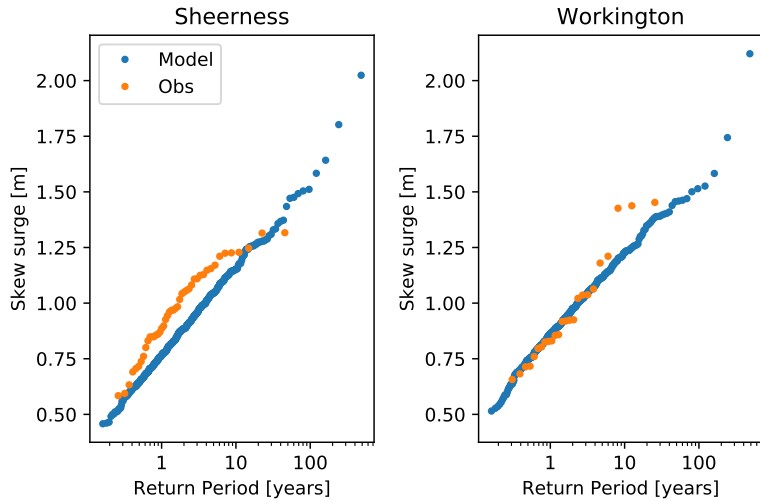

**Figure 1.** Empirical return level plots of skew surge, comparing simulated and observational (tide-gauge) data at Sheerness and Workington.

Figure 1 shows excellent model vs observations agreement for the two sites illustrated, and even these two sites alone illustrate that our modelling system is able to simulate unprecedented skew surge events (i.e. events of a magnitude not found in the tide-gauge record). However, the quality of agreement shown in Fig. 1 is not exhibited everywhere. Empirical return
level plots of skew surge for a set of 44 tide gauge locations around the UK coastline are shown in the appendix in Fig. B1. This gives a qualitative, visual sense of the realism of the model in terms of the simulated extremes. The excellent agreement at Sheerness and Workington can be contrasted with the poor agreement at, for example, Newlyn or Aberdeen, where the simulated extremes are negatively biased relative to the corresponding observations. We show below that the simulation may nevertheless be able to add value to estimations of unprecedented events, even where a bias exists.

**4.1   Quantitative evaluation of simulation of extremes**

To make some quantification of the realism of the simulated extremes, we used the statistical models described in §3.3 to fit the simulated extremes. We fitted a GEV model (§3.3) to the simulated annual maxima using the MLE method (§3.3). We fitted the model pointwise (that is, for each tide gauge we fitted independently at a model grid cell closest to the tide-gauge of that site; this model grid cell is taken to represent that site). This gives a spatial distribution of diagnosed parameters. We find
excellent agreement between the simulation-based and tide-gauge-based location and scale parameters (Fig. 2). We also find a surprisingly good correlation between the spatial distribution of simulation-diagnosed shape parameters and the corresponding spatial distribution of shape parameters diagnosed by CFB2018. Pearson's r for the shape parameter correlation is 0.72 when we use a GEV fit to the simulated annual maxima, and 0.86 when we use a GPD fit to the simulated peaks over a threshold (see §4.2).



**Figure 2.** Comparison of simulation-based and observation-based skew surge extreme value distribution parameters. (a): Location parameter. (b) GEV scale parameter ($\sigma$). (c, d): Shape parameter. The correlation seen in all panels shows that the model successfully simulates the observed spatial variations in the extremes. For full details see main text.





A detailed description of Fig. 2 panels (a,b,c) follows. A complete description of panel (d) is deferred to §4.2. The corre-
lation in all panels shows that the model successfully simulates the observed spatial variations in the skew surge extremes. In
particular, good representation of the scale parameter at each site is important because this means that the temporal variability
is well-simulated at that site. The absolute size of the scale parameter is significant, so we include zero in the Y-axis of panel
(b). A scale parameter of zero would indicate no inter-annual variation in the extremes.

If we were to base our assessment on, for example, SWL relative to local Chart Datum (instead of skew surge), then the
absolute value of the location parameter would have no particular significance: it would depend on a local offset. However, for
skew surge the absolute value of the location parameter does have a significance: it represents a hypothetical absence of any
atmospheric effects. For that reason we also include zero in the Y-axis of panel (a). A location parameter of zero would indicate
no atmospheric effect on sea levels.

For all sites shown in Fig. 2, we obtained sufficient information regarding the CFB fit to the observations (pers. comm: Jenny
Sansom, by email) to enable us to evaluate their GEV scale parameters (panel (b)). Their shape parameters (panel (c) and (d))
can be read from their figure E.1. For the location parameter (panel (a)) we used our own more crude fit to the tide-gauge
annual maxima. We confirmed this crude fit against additional CFB information at a sample of nine sites. The crude fit was
only used to estimate the observational location parameter.

Consideration of Fig. 2(a) shows that the simulation-diagnosed location parameters are in general slightly low compared to
our crude estimate from the tide gauge data. At some sites this may be associated with locally poor representation of the details
of the coast and bathymetry around the tide gauge due to the surge model resolution. However, the scale parameter (panel (b))
is generally in very good agreement with the CFB2018 results. This is reassuring because it indicates that the simulation is
doing a good job of capturing the variability in the extremes (scale parameter), even though it shows an overall bias in the
extremes (location parameter).

    Panel (c) has three main features:

1.  The shape parameters diagnosed from the simulation are well correlated with the CFB2018 shape parameters. This
    strong correlation between the two spatial patterns of shape parameter diagnosed from independent sources (i.e. our
    model simulation and the tide-gauge data) is remarkable. It both supports the spatial pattern of the shape parameter as
a real, physically-determined phenomenon (as opposed to a statistical artefact), and gives further credibility to both the
    CFB2018 approach and our model. The authors are not aware of any previous work in which the spatial pattern of skew-
    surge shape parameter diagnosed from a simulation based on a free-running climate model has been shown to correlate
    well with the corresponding pattern diagnosed from observations.

2.  The spread of the shape parameters diagnosed by MLE fit (i.e. without constraint) to annual maxima from the simulation
is comparable to the spread of the CFB2018 shape parameters diagnosed by PMLE (i.e. with constraint), which in turn is
    similar to the spread of the subjective prior used by CFB2018. This again suggests that a long climate model simulation
    may be useful in constraining the shape parameters.





3. The fitted shape parameters for the simulation are more negative than the CFB2018 shape parameters. We return to this in §4.2.

The sites in Fig. 2 follow a clockwise orbit of the UK mainland coast starting at Newlyn in the south west, with the addition of St Mary's (Isles of Scilly), Port Erin (Isle of Man), Stornoway (Outer Hebrides), Lerwick (Shetland Isles) and Portrush (Northern Ireland). The sites are shown in Fig. A1(b) in the appendix.

## 4.2   Shape parameter

We return now to the fitted shape parameters for the simulation, which are more negative than the CFB2018 shape parameters.
This is important because uncertainty in estimating unprecedented events from observational records using MLE is dominated by uncertainty in the shape parameter (see appendix C). This suggests that the shape parameter is the aspect where model simulations, with their long record lengths, may be able to help. We studied the negativity in several ways, with results which are shown in Fig. 3 and Fig. 2 panels (c) and (d). Simulation results shown in Fig. 2 panels (a to c) are from a GEV fit to the simulated annual maxima, whereas CFB2018 results are from a GPD fit to the observations. To eliminate this potential source
of difference, we also made a GPD fit to the simulation (Fig. (2) panel (d) and Fig. 3 line (C)). We applied the same treatment to simulations as was used by CFB2018 in order to make a like-for-like comparison. We did not apply a prior to produce the simulation shape parameters shown in Fig. 2 and Fig. 3 line (C).





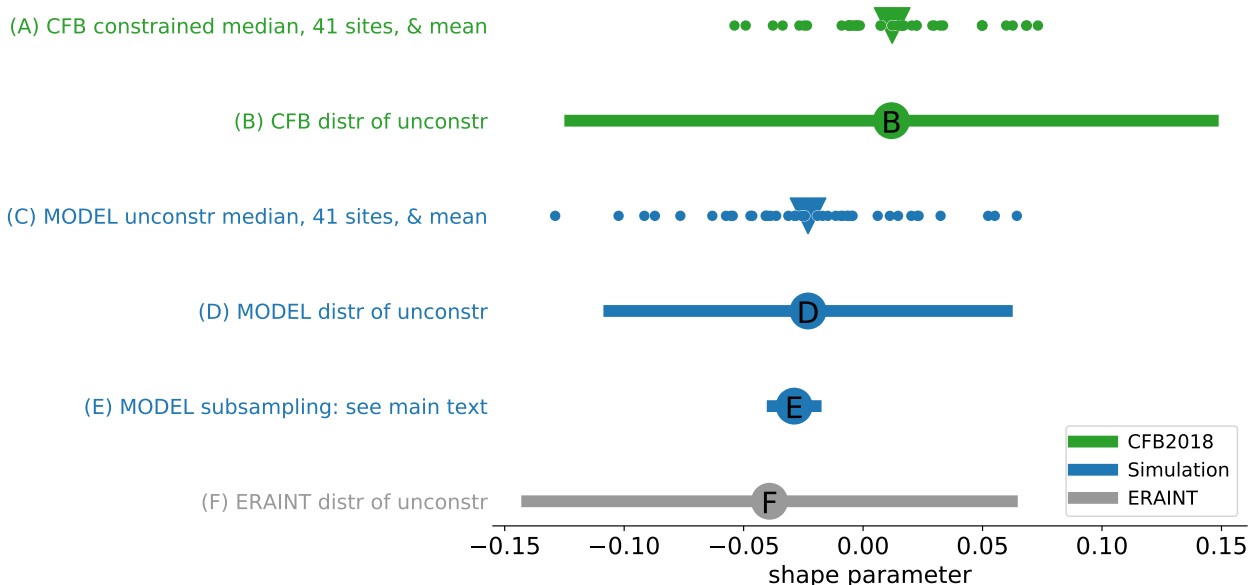

**Figure 3.** Further results related to the model vs CFB2018 shape parameter difference. Each line shows (X-axis; dimensionless) a distribution of GPD shape parameters, or a derived quantity such as the mean of several shape parameters. "unconstr" = Unconstrained. For full details see the main text.

Details of Fig. 3 follow. Lines (A) and (B) show shape parameter results from CFB2018. They used a GPD fit to skew surges exceeding a threshold. They used an extremal index (Tawn, 1992; Batstone et al., 2013) to accommodate dependence in the time-series. To compensate for the short observational record lengths, they used a prior to constrain the shape parameter (see §3.3). The prior (or "penalty function") in turn was chosen by expert judgement informed by unconstrained GPD fits to the tide-gauge data. 14 different thresholds were tested, and results for each site and each threshold were pooled to form a distribution of unconstrained shape parameters. This distribution is shown in line (B). The green line shows the range (characterised by two standard deviations either side of the mean). The filled disc labelled "B" shows the mean. The CFB2018 prior was taken to be a normal with the same mean but half the standard deviation. For each site, the finalised (constrained) shape parameter diagnosed by CFB2018 was chosen as the median of the PMLE results of the 14 different thresholds. These finalised shape parameters, one for each of 41 sites, are shown by the dots in line (A). (This same information is contained in Fig. 2 panels (c) and (d)). The mean of these 41 shapes is shown by the filled triangle. We applied a similar approach (but without the prior) to our simulated skew surges to give the results shown in line (C). The Pearson's r correlation between the model-diagnosed shape parameters using GPD fits (line C) and the CFB2018 shape parameters (line A) is 0.86. Owing to the much greater record length of the simulation, we did not need to apply any constraint to obtain the data on line (C).





The data of line (C) vs the data of line (A) show our most like-for-like simulation-vs-CFB2018 shape parameter comparison, and are also presented in Fig. 2 panel (d). That panel also shows our estimate of the uncertainty in the CFB2018 shape parameters, expressed as a 95% confidence interval. This estimate is based on the standard error of the CFB2018 fit at the 95%

threshold (pers. comm: Jenny Sansom, by email). It is not straightforward to estimate the uncertainty of the CFB2018 shape parameters owing to their use of a median over results based on different thresholds ranging from 90% to 99%, but we suggest that the uncertainty in their 95% threshold result is representative.

Line (D) represents the full distribution of shape parameters diagnosed by GPD fit to the 483-year HadGEM3-driven model simulation. The blue line shows the range (characterised by two standard deviations either side of the mean). The range (as

in line B) comes from variations in site and threshold used. The filled disc labelled "D" shows the mean. Thus D (model) corresponds to B (tide gauge). Clearly the 483-year model-diagnosed shapes are more negative than those derived from the shorter tide gauge data.

Given the need for some kind of constraint on the shape parameter when fitting observational records, use of shape parameters from a long simulation holds the promise of reducing uncertainties without the subjectivity of a prior. Thus, the

more-negative shape parameters diagnosed by fitting the model data are, potentially, our most important finding, but further work is required to better understand the causes of this negativity. On one hand, it could be that limitations in the realism of either the atmospheric or the coastal shelf modelling distort the distributional tail relative to the real world. On the other hand, it could be that the physically-based model simulation gives better guidance on the distributional tail of the atmospheric storms which drive surges than does a statistical fit to the relatively short observational record of the surges themselves. In

favour of the simulation, we can say that the emergence of realistic long-period natural variability in climate model simulations suggests their suitability for generating samples outside the observational record length. If it could be shown that the long-period variability in the simulation envelopes the observational results, this would give much stronger support to the use of the simulation.

Could it be, then, that if the simulation were sub-sampled in shorter periods to match the tide gauge record lengths, the value

of a new metric (call it $D'$, the mean of the distribution of shape parameters diagnosed by GPD fit to the shorter sub-sampled HadGEM3-driven model simulation) would vary substantially so as to sometimes include values as large as "B"? To answer this question we sub-sampled the model many times to give a distribution of $D'$. This distribution is represented by line (E): the blue line shows the range of values of $D'$ (characterised by two standard deviations either side of the mean; this range comes from random variations which we have introduced into the start time of the sub-samples, so that each sub-sample represents

a randomly-chosen different "era" of the simulation) and the filled blue disc labelled "E" shows the mean value of $D'$. It is clear that this distribution *does not* include values as large as "B", meaning that the *apparent positive* shape parameter bias of the tide gauge results (B) relative to the model results (D) is *not* simply a "sampling error" associated with the shorter record lengths of the tide gauges, but rather a *real negative* bias in the model shape parameters relative to the tide gauges.

However, line (F) shows the distribution of unconstrained shape parameters diagnosed from a 29-year CS3 run forced by

atmospheric surface wind and pressure based on the ERA-interim atmospheric reanalysis (Dee et al., 2011) that has been downscaled with the Swedish Meteorological and Hydrological Institute (SMHI) Rossby Centre regional atmospheric model





(RCA4) as part of the Euro-CORDEX experiment (Jacob et al., 2014). This distribution shows at least as much negative bias as the HadGEM3-driven model simulation, even though the ERA reanalyses are widely viewed as the gold standard in terms of representing the storminess of the real atmosphere. The foregoing suggests that the negative bias is due to the limitations

of the CS3 surge model, which is common to both the HadGEM3-driven and the ERA-interim-driven results, and that the HadGEM3 simulation of storminess is comparable (at least by this metric) to the ERA reanalysis. In short, the atmospheric model is adequate, but we need a better surge model.

Further shape parameter results are shown in appendix D.

Very recently, Horsburgh et al. (2021) have simulated surge events which are higher than the CFB2018 best estimate of

1000-year return level. This finding is not contradictory to ours. Horsburgh et al. (2021) seek to identify unprecedented events which are *possible* in the present-day climate, without seeking to quantify their probability. Our focus is on improving the quantification of the probability of unprecedented events.

## 5   Sheerness case-study experiments

In view of the societal and economic importance of the Thames estuary, we further investigate the behaviour at Sheerness.

### 5.1   Tide-Surge timing

Figure 4 shows results of experiments making small (less than 24 hours) timing shifts in the phase relationship between the atmospheric forcing and the tide. We chose a spring tide and shifted the timing of the event relative to the tide. The curve labelled "SWL 0" shows the SWL of the shift which gives the maximum skew surge. The curve labelled "SWL 4" shows the SWL when the event is shifted 4 hours later relative to the tide.. The skew surge on the initial high tide (about nominal hour

24 in the figure) is reduced by about 1.2 metres. The overall maximum SWL (which occurs on the next high tide in the shifted case) is reduced by about 0.8 metres.



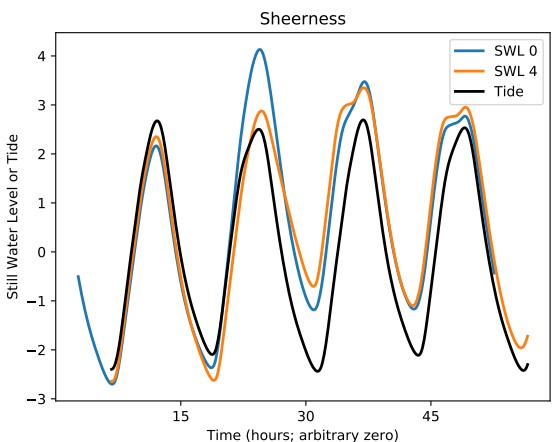

**Figure 4.** A simulated extreme event on a spring tide showing the effect of a shift of 4 hours in the timing of the event relative to the tide.

Clearly, a potentially extreme event may not be realised as an extreme SWL if does not happen to be in a conducive timing relationship with the tide. From a coastal defence viewpoint this is good, as it reduces the number of extreme SWLs which are realised. But from the viewpoint of identifying extreme events in a long model simulation it is a nuisance, because it can mean that potentially extreme events are hidden. To overcome this we performed a further simulation with the surge model in surge-only mode (see §6.1). In this mode no astronomical tides are included, and therefore all potentially extreme atmospheric events are realised as a surge.

## 5.2 Skew Surge/Tide dependence at Sheerness

Work by Williams et al. (2016) has shown that any dependence of skew surge on predicted high water cannot be readily quantified in the observational record, due to the dominance (in the record) of the variability of atmospheric storms. This conclusion has lead to the exploitation of an assumed independence of skew surge and predicted high water as part of the effort to estimate present-day still water return levels — the so-called skew surge joint probability method which is used by CFB2018 (although they do note that such independence is not applicable everywhere).

This independence can be tested in model simulations, by repeating the same atmospheric storm in different astronomical tidal conditions – for example at spring and neap tide. Williams et al. (2016) perform four experiments of this kind (see their supplementary material) using reanalysed real-world storm data. We extended that work using 16 of the most extreme forcing events (in the sense that they create an extreme surge at Sheerness) from our HadGEM3-GC3-MM control simulation. Results are shown in appendix E. Here we give an example of a single event.

The largest skew surge event at Sheerness in the HadGEM3-GC3-MM simulation happened to arrive on a neap tide. Figure 5 shows that when this event was moved to a spring tide, the skew surge was significantly attenuated, from about 2 metres in the



case of the neap tide to about 1.63 metres in the case of the spring tide. Williams at al.(2016, their supplementary material S5) also found attenuation at Sheerness in model simulations of four events.

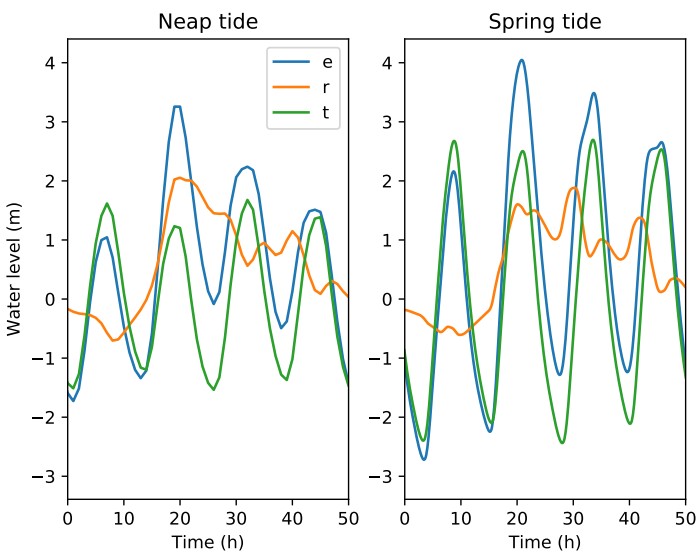

**Figure 5.** Left panel shows the largest skew surge in the HadGEM3-GC3-MM 483 year surge-and-tide simulation, which happened to arrive on a neap tide. Right panel shows the same atmospheric forcing applied to a spring tide. It can be seen that the realised skew-surge is dependent on the tide in this case. Key: e: still water elevation. t: astronomical tide. r: residual (i.e. e-t). X-axis shows time in hours with arbitrary zero.

Further skew-surge/tide dependence results are shown in appendix E.

## 6   Sheerness: comparison of the most extreme simulated events with reconstructions of the 1953 event

### 6.1   Sheerness: Surge-only simulations

Our surge-only simulations are motivated by the sensitivity shown in Figs. 4 and 5, and discussed in §5. Using a numerical coastal shelf model it is possible to artificially eliminate the effect of the astronomical tide to create a surge-only simulation. Thus, issues of the timing relationship between surge and tide are eliminated in a surge-only simulation and so the sensitivity is avoided. This makes surge-only simulations well suited to comparing different sets of atmospheric forcing in terms of their
surge-creation potential for a given location.

Figure 6 shows time series of water level at Sheerness for 16 events from our HadGEM3-GC3-MM surge-only simulation, in each case compared with a surge-only simulation driven by atmospheric data from a reconstruction (pers. comm: Erik van Meijgaard, by email) of the 1953 storm using the KNMI/DMI limited area model RACMO (van Meijgaard et al., cited 2020).


We selected first the largest 8 events in terms of the maximum value of the surge which they produced in surge-only mode.

Compared to these events, the 1953 surge-only simulation has a conspicuous long duration. So we also sought events of long duration by convolving the surge only time series with the kernel shown in Fig. 6 panel i. Then we identified the 8 largest maxima in the convolved signal. All 16 events are independent (all separated from each other by at least a year).

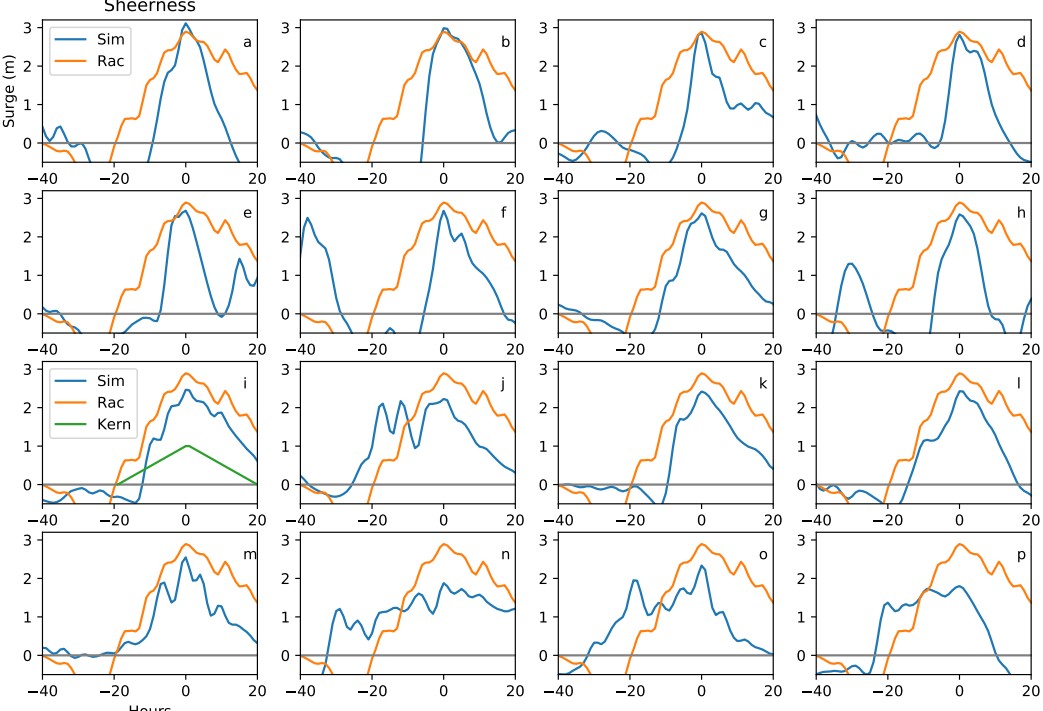

**Figure 6.** 16 events from the HadGEM3-GC3-MM surge-only simulation ("Sim"), in each case compared with the RACMO-driven surge only simulation ("Rac"). Panels a to h show the 8 largest independent surge-only events. Panels i to p show events which are both large and have substantial duration (the original time series before convolution is shown). X-axis is time in hours with arbitrary zero.

This shows that in the 483-year surge-only simulation

1. No simulated event exceeded the 1953 reconstruction in terms of both maximum surge *and* duration.

2. Two simulated events exceeded the 1953 reconstruction in terms of maximum surge, and *several more were comparable*.

3. Several simulated events were of comparable duration to the 1953 reconstruction, but exhibited a smaller maximum surge.



## 6.2 Sheerness: Surge and Tide simulations

Having used the surge-only mode to identify 16 potentially-extreme events in the HadGEM3-GC3-MM simulation, for each

event we experimented with adjusting the timing of the event in a surge-and-tide simulation to maximise the skew surge realised. We did this twice: once for a spring tide and once for a neap tide. Figure 7 shows (bar "S") the overall (i.e. over all 16 events) maximum skew surge realised on a spring tide and similarly the overall maximum skew surge realised on a neap tide (bar "N"). Figure 7 also shows (bar "H") the maximum skew surge realised in the original HadGEM3-GC3-MM surge-and-tide simulation, in which the timings were not artificially adjusted, so that the surge/tide phase relationship was essentially

random (as in the real world). For reference an extreme (entirely artificial) case is shown (bar "Z") in which no tidal forcing is included (see §6.1). In reality, of course, the tide is always present. Wadey et al. (2015) tabulate estimates of high water level at Sheerness for the 1953 event from four different sources. They also give a best estimate of 4.74 metres, and a corresponding best-estimate skew surge of 2.16 metres. This implies an astronomical tide of $4.74 - 2.16 = 2.58$ metres. Thus, to obtain the four skew surge estimates labelled "W" in Fig. 7 , we subtract this tide from each of their four tabulated estimates of high water

level.





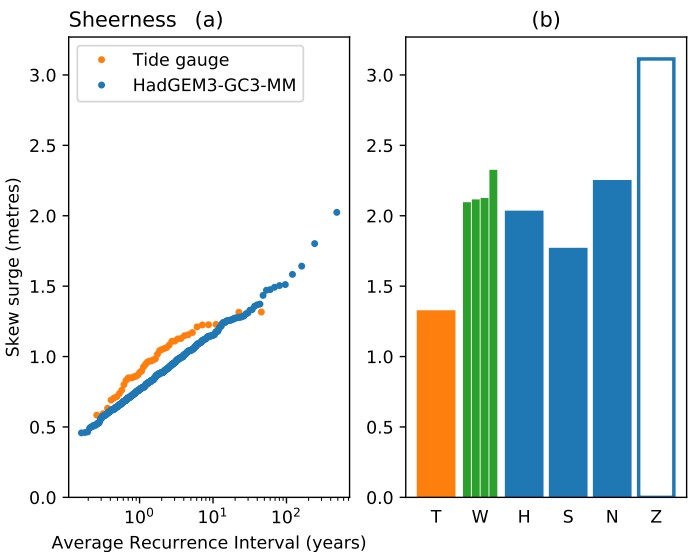

**Figure 7.** Observed/estimated (in orange) and modelled (in blue) skew surges at Sheerness. (a): Empirical return level plot showing annual maxima of skew surge from tide gauge data as used in CFB2018 (orange) and from 483-year model simulation (blue). Years with less than 75% of available data are excluded from the tide gauge data analysis. (b): Skew surge maxima. Key: T(ide): Tide gauge (max from panel (a)). W(adey): Data from four different sources for the 1953 event, as tabulated by Wadey et al. (2015). H(adGEM3): Model max skew. Phase relationship between atmospheric events and astronomical tide is essentially random over the 483 year simulation. S(pring): Model max skew when events are artificially shifted to coincide with Spring tide. N(eap): Model max skew when events are artificially shifted to coincide with Neap tide. Z(ero): Model max skew when astronomical tides are excluded (surge-only simulation: "Zero tide").

Figure 7 shows that the strongest atmospheric forcing in the model simulation can produce a skew surge which is comparable to estimates of the 1953 skew surge at Sheerness. The largest skew surge (bar "H") in the simulation in which the timings of atmospheric forcing and tides are essentially random, lies just below the range of skew surge estimates based on data tabulated by Wadey et al. (2015). The largest spring-tide skew surge (bar "S", when the timings of atmospheric forcing are adjusted

so that the atmospheric events coincide with a spring tide) is smaller than the observational estimates, due to the surge-tide interaction at this site. The largest neap-tide skew surge (bar "N", when the timings of atmospheric forcing are adjusted so that the atmospheric events coincide with a neap tide) lies within the range of skew surge estimates based on data tabulated by Wadey et al. (2015).

We do not attempt to explicitly quantify a return period of the 1953 SWL at Sheerness because an evaluation of the probabil-

ity distribution of SWL requires a convolution of the distributions of skew surge and of tide (see §3.3). CS3, in common with other shelf models, is known to exhibit tidal errors, typically under-predicting the range. However, the fact that the 483-year surge-only simulation produces more than one event of comparable magnitude to the simulated 1953 event suggests that the



return period of the 1953 atmospheric forcing is less than 483 years, in so far as the model is realistic. Wadey et al. (2015) suggest a return period of 429 years for the surge event at Sheerness.

## 7  Summary and Conclusions

HadGEM3-GC3-MM is a state-of-the-art global climate model of the CMIP6 generation. Modifications including the ENDGame revision to the dynamical core have been shown to increase synoptic variability (Williams et al., 2015), improving the representation of the storm tracks compared to HadGEM2-AO (the Hadley Centre model which contributed to CMIP5). We have shown that a 483-year control simulation of HadGEM3-GC3-MM, in combination with a barotropic storm surge model of the north west European coastal shelf, is capable of directly simulating realistic extreme storm surges for some sites around the UK coastline, as evaluated against observations (Fig. 1). In particular, our modelling system simulates several surge events at Sheerness (on the Thames Estuary) which are comparable to best estimates of the catastrophic 1953 storm (Figs. 6 and 7).

We extend the skew surge–tide dependence results of Williams et al. (2016). Our simulations suggest that skew surge–tide dependence can have a substantial effect on the most extreme surges at Sheerness (Fig. 7 and appendix E).

Furthermore, around the whole of the UK coastline we find that the spatial pattern of variations in the three parameters which describe the extreme tail of the storm surge distribution is very well reproduced by the simulation (Fig. 2). In particular, the observed spatial variations in the shape parameter are reproduced by the simulation. This is important because

- it gives further credibility to both diagnoses of the spatial variations

- the shape parameter is the main source of uncertainty in estimates of unprecedented events (appendix C)

- the length of the simulation (much greater than the length of the observational record) helps to constrain the shape parameter with less subjectivity (§4.1).

A typical simulated shape parameter for an individual site is more negative than (but within the uncertainty of) that diagnosed by CFB2018 (Fig. 2 panel (d)). This negativity arises at a wide spread of sites. Such spatial uniformity of the negativity strongly suggets an underlying difference rather than a chance/sampling difference. Sub-sampling the simulation with sample sizes matching the tide gauge record lengths supports that suggestion and shows that our model shape parameters are biased low relative to those diagnosed from tide-gauge observations. However, that is also the case when our surge model is driven by a good quality atmospheric reanalysis, suggesting that the bias comes from shortcomings at the surge modelling stage rather than the atmospheric forcing.

We conclude, then, that our atmospheric model, HadGEM3-GC3-MM, has the potential to help constrain estimates of unprecedented UK storm surges, but that improvements at the surge modelling stage are required.

*Data availability.*  The tide gauge data used in the CFB2018 report are available to download from the National Tidal and Sea Level Facility (ntslf.org). The CFB2018 shape parameters can be read from their figure E.1 (Environment Agency, 2018). The CFB2018 GEV scale

parameters as shown in Fig. 2(b), in metres, (for sites Newlyn... Portrush in the order shown on the X-axis of Fig. 2) are:

0.0835, 0.0733, 0.0847, 0.1031, 0.1369, 0.1790, 0.1574, 0.1484, 0.1140, 0.0940, 0.1639, 0.1063, 0.1161, 0.1305, 0.1228, 0.1815, 0.1593,

485  0.1358, 0.1757, 0.1490, 0.1471, 0.1202, 0.0955, 0.1368, 0.0684, 0.0913, 0.0922, 0.0966, 0.1049, 0.1178, 0.1333, 0.1618, 0.1748, 0.2153,

0.1715, 0.1629, 0.1557, 0.1091, 0.0985, 0.0937, 0.0948, 0.1201, 0.0938, 0.1273

Simulated sea levels at the tide gauge sites as used in our analysis are available from the first author on request. All of the analysis was undertaken using the open source languages R and Python.

## Appendix A:  Surge model grid and tide gauge locations

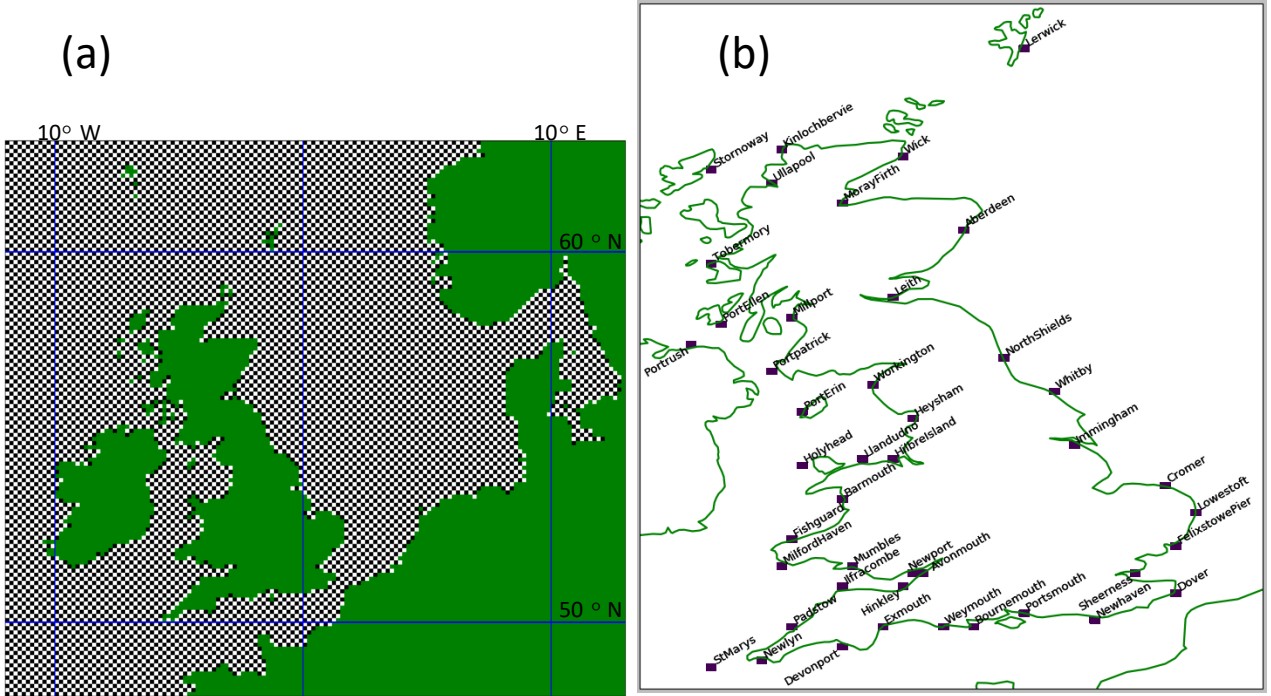

**Figure A1.** (a) Domain and grid of the CS3 coastal shelf model. Grid size is 1/9 degree in latitude and 1/6 degree in longitude, which results in near-square grid cells at the latitude of the UK. (b) Tide gauge locations.

## Appendix B:  Empirical return level plots for UK tide gauges

Figure B1 shows empirical return level plots for 44 tide gauge locations around the UK. For simplicity we use annual maxima only. The observational annual maxima are limited to years in which the tide gauge data is at least 75 % complete. Plotting positions are evaluated using the Weibull formula (Weibull, 1939). Working from left to right along the rows (and then downwards through the rows as in reading), the sequence of plots follows the clockwise sequence of Fig. 2 as described in §4.1.



Natural Hazards
and Earth System
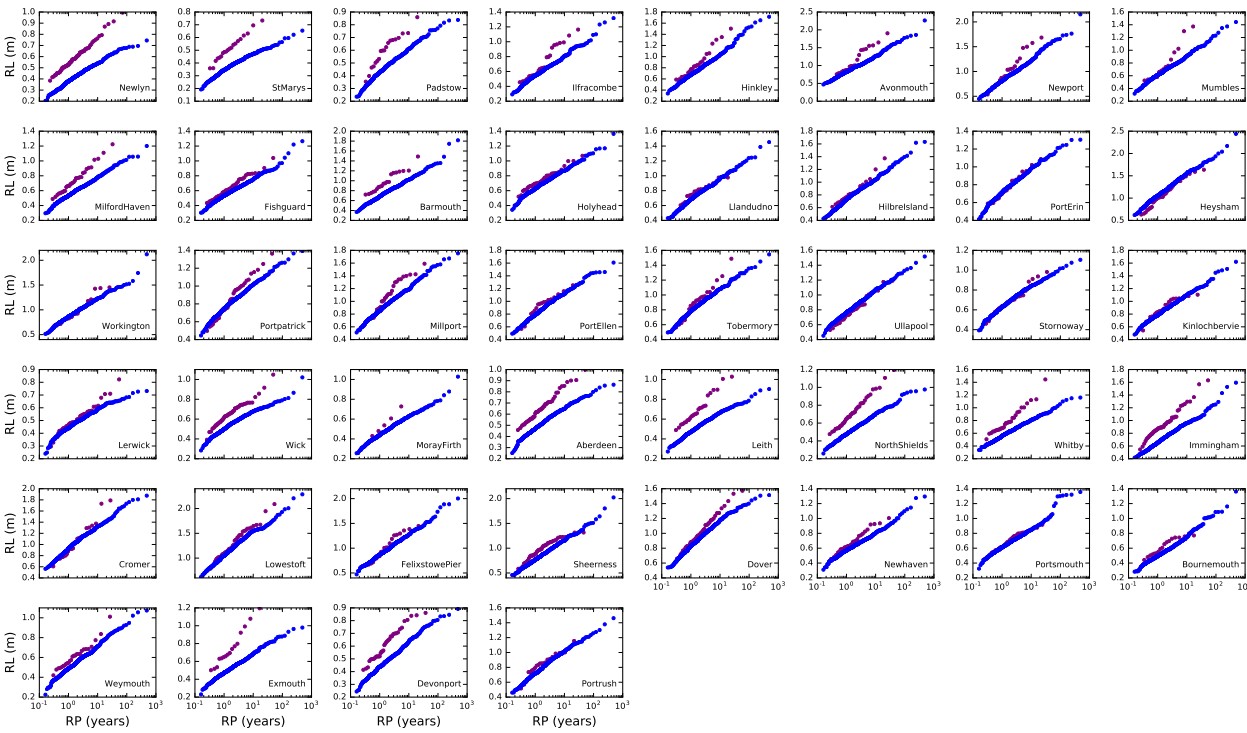

**Figure B1.** Empirical return level plots for 44 UK tide gauges. Blue shows observational (tide-gauge) data. Purple shows data from the 483-year HadGEM3-GC3-MM simulation.

It can be seen that at some locations the model produces a plausible simulation of the observed return level plot and a plausible extrapolation of the return level plot to return periods outside of the observational record. This is discussed further in §4.

### Appendix C:  Shape-parameter uncertainty dominance

For short record lengths, unconstrained maximum-likelihood estimation is known to give "noisy" and implausible shape pa-
rameters (Coles and Dixon, 1999; Martins and Stedinger, 2000), see also §3.3. We illustrate this in Fig. C1 with a GEV fit to tide-gauge data.

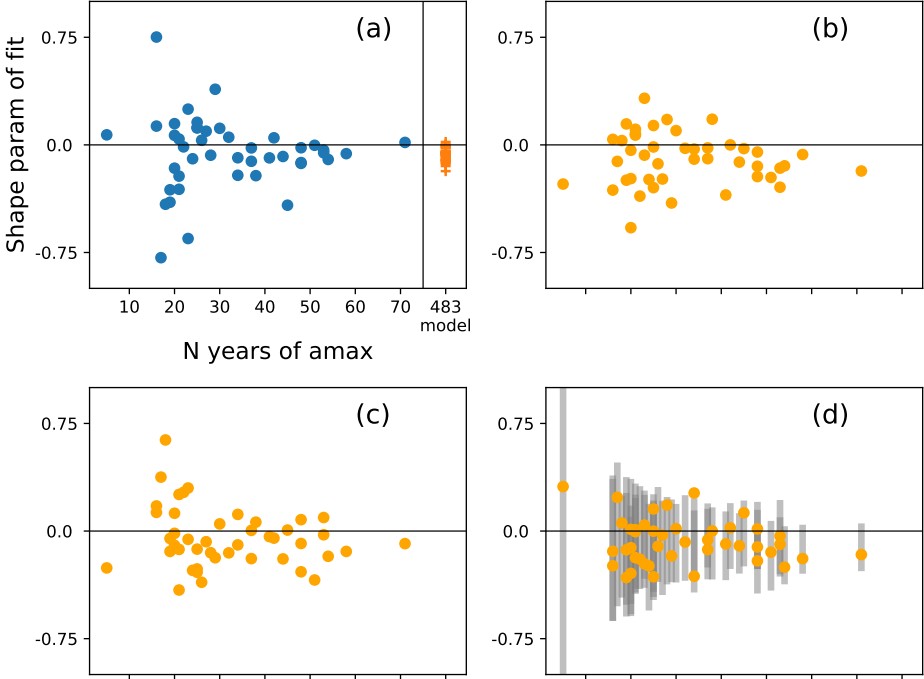

**Figure C1.** Short record lengths lead to noisy MLE shape-parameter estimates. (a) Shape parameter of GEV fit to tide gauge data against the number of annual maxima fitted (blue dots, one for each tide gauge), and shape parameter of GEV fit to model data (orange crosses, one for each tide gauge, all 483 years). Note that the range of fitted shape parameters reduces ("tapers") as record length increases. (b) Model data for each port is cut down to a (random) sub-sample having the same length as the observational record at that port and then fitted in the same way as the observations. A similar tapering pattern emerges. (c) as (b) but a different random sub-sample. (d) as (b) but a different random sub-sample. Also shown (grey) is the 5 to 95 percentile range from one hundred such sub-samples at each port.

In similar plots for the location and for the scale parameter, no such tapering is exhibited In this illustration, tide gauge records of length greater than about 40 years have a fitted shape parameter which is within the range of the model fitted shape parameters; only in short records are large positive or negative shape parameters found. Figure C1 (b), (c) and (d) confirm that

the tapering is a result of record length.

Associated with this, for observational record lengths, the uncertainty in the shape parameter dominates the uncertainty in inferred return levels for long return periods. This is illustrated in Fig. C2.





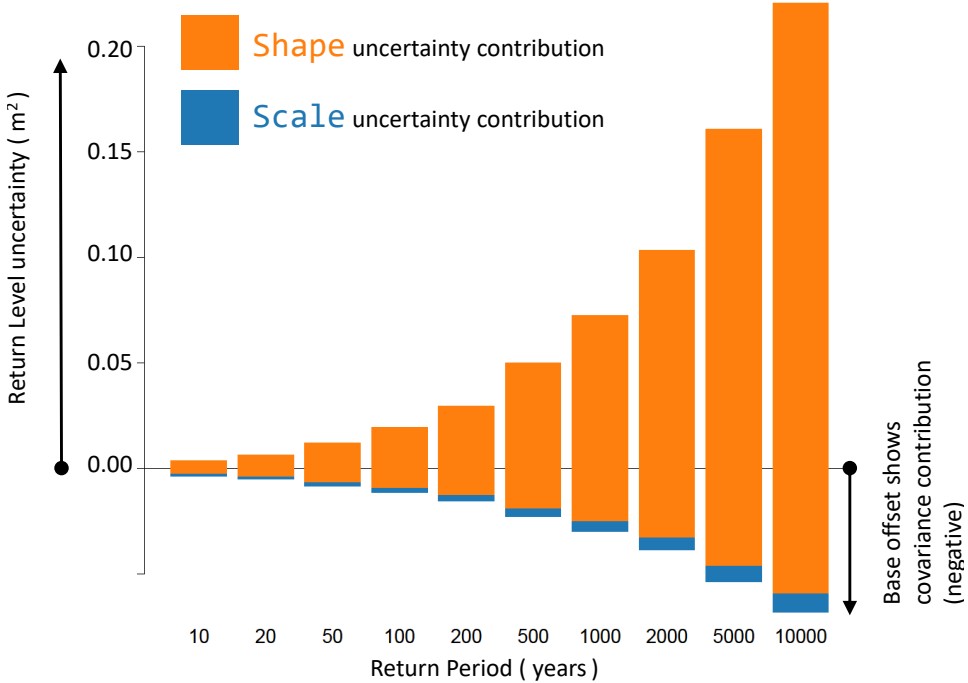

**Figure C2.** The uncertainty (variance) in different return levels is partitioned into contributions from uncertainty in the shape parameter, uncertainty in the scale parameter, and a negative contribution from the covariance of these two parameters, shown as an offset to the base of the bars. Uncertainty in the shape parameter becomes dominant at long return periods.

Figure C2 shows the sources of uncertainty in return level for ten different return periods. The data were constructed as follows. We took representative shape and scale parameters (we used the CFB2018 parameters for skew surge at Sheerness)
and a representative record length of 45 years. We simulated 45 years of 94 % threshold exceedance data by inverse transform sampling. We estimated the (GPD) parameters of the sample by maximum likelihood estimation without any prior constraint. For each of ten return levels, we estimated the uncertainty using the delta method (e.g. Coles, 2001) as follows.

$$\mathrm{Var}(R) = \frac{\partial R}{\partial \widetilde{\sigma}}\mathrm{Var}(\widetilde{\sigma})\frac{\partial R}{\partial \widetilde{\sigma}} + \frac{\partial R}{\partial \xi}\mathrm{Var}(\xi)\frac{\partial R}{\partial \xi} + 2\frac{\partial R}{\partial \widetilde{\sigma}}\mathrm{Cov}(\widetilde{\sigma},\xi)\frac{\partial R}{\partial \xi} \qquad (C1)$$

where $R$ is return level, $\widetilde{\sigma}$ is the GPD scale parameter, and $\xi$ is the shape parameter. The variance and covariance terms, which
are determined from the curvature of the likelihood surface, are evaluated during the likelihood maximisation routine. The three terms on the right-hand side of equation C1 are the contributions to the return level uncertainty from the GPD scale parameter uncertainty, the shape parameter uncertainty, and the covariance of the two parameters, respectively. To increase confidence in the uncertainty estimates we repeated the sampling many times and averaged over each contribution. A further contribution to uncertainty is the choice of threshold, but this contribution is usually found to be small (Coles, 2001) and is neglected
here. We tested some alternative approaches (not shown here), for example fitting a GEVD to the annual maxima instead of GPD to POT. The essential result — the dominance of the shape parameter uncertainty — is robust and was not affected by





the use of alternative approaches. The result holds for all of the nine locations that we tried: Newlyn, Fishguard, Holyhead, Stornoway, Lerwick, Aberdeen, Cromer, Lowestoft and Sheerness. In each case we simulated a record length corresponding to the available tide gauge data for that location.

**Appendix D:  Further shape parameter results**

Figure 3 in the main text shows results of fitting the GPD distribution to peaks over a threshold. Here, in Fig. D1, we extend that figure to include results of fitting the GEV distribution to annual maxima.

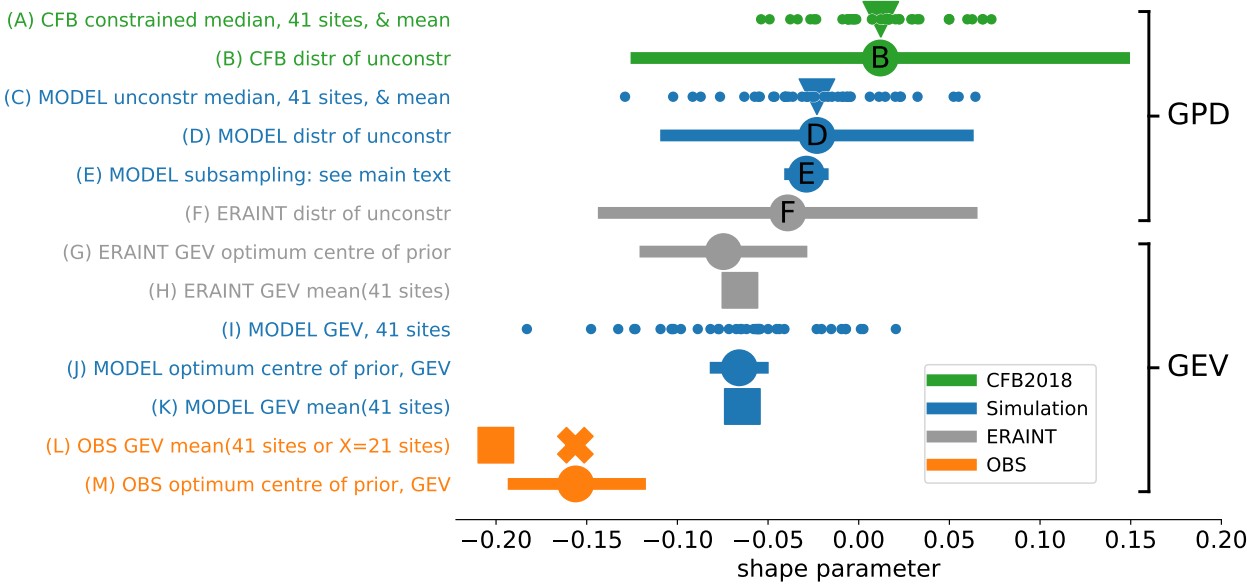

**Figure D1.** Further shape parameter results. (A to F): diagnosed by GPD fit to POT as Fig. 3. (G to M): diagnosed by GEV fit to annual maxima.

To probe the model–tide gauge shape parameter bias further we performed some more experiments. We made an unconstrained GEV fit to tide-gauge annual maxima of skew surge at each of the 41 sites. Owing to the short record length the results are very noisy and include some implausible values. Nevertheless the mean of these 41 results is shown by the filled square on line (L). The mean for the 21 sites with the longest record lenghts is also shown, by the cross on line (L). For comparison, the mean of the GEV fit to simulated data (41 sites) is shown in line (K). The GEV fit to simulated data for each of the 41 sites is shown in line (I) (i.e. the square on line (K) is the mean of the points on line (I)). To compensate for the short observational record length we experimented with applying a prior to the shape parameter of the GEV fit to the observed annual maxima of skew surge. Following CFB2018, we used a normal prior with a standard deviation of 0.0343, but we varied the centre (i.e. the





mean) of the prior. For each site, we produced a maximum likelihood fit by maximising the log-likelihood in the usual way. We summed log-likelihoods over all sites to give an overall log-likelihood for that value of the centre of the prior. The value of the centre of the prior which maximised the log-likelihood is shown by the disc in line (M). Standard techniques (Coles, 2001) enable us to identify a 95 % confidence interval, shown by the solid straight line in line (M). We applied the same approach to
the simulated skew surges to give the results shown in line (J).

We did some further shape parameter evaluations with surges generated by CS3 driven by the ERA-interim atmospheric reanalysis (Dee et al., 2011) that has been downscaled with the Swedish Meteorological and Hydrological Institute (SMHI) Rossby Centre regional atmospheric model (RCA4) as part of the Euro-CORDEX experiment (Jacob et al., 2014). The ERA-interim GEV-mean (corresponding to the square in line (L)) is shown in line (H). The GEV-optimum centre of prior (corre-
sponding to line (M)) is shown in line (G). In all cases, our ERA-interim-based shape parameter results are in better agreement with our model results than with the CFB shape parameter results. As discussed in the main text, this suggests that the bias is not a result of any deficiency in the climate-model compared to the ERA-interim reanalysis, and that it is more likely arising from the continental shelf modelling stage.

The GEV fits to skew surge data give more negative shape parameter results than GPD fits. This is surprising since both
methods (GEV and GPD) are asymptotically unbiased (Coles, 2001). We further experimented with fitting to some entirely artificial data (generated pseudo-random numbers) of comparable size to the simulations. We did not find any evidence of a consistent negative shape bias in the MLE when using a GEV compared to a GPD fit for any distribution of pseudo-random numbers that we tested (including Gumbel, normal, and GEV with +ve and -ve shape parameter). Thus the GEV vs GPD bias may be hinting at a departure from the conditions which are required for accurate statistical modelling of the extremes.

**Appendix E:  Further skew-surge/tide dependence results at Sheerness**

For each of the 16 extreme events shown in Fig. 6 we re-ran the simulation, adjusting the timing such that the event coincided with a spring and then a neap tide. In each case we also made small timing adjustments to realise the maximum skew surge. In all cases the skew surge was attenuated on the spring tide relative to the neap tide. The size of the skew surge, and the spring-neap difference in the skew surge are shown in Fig. E1.

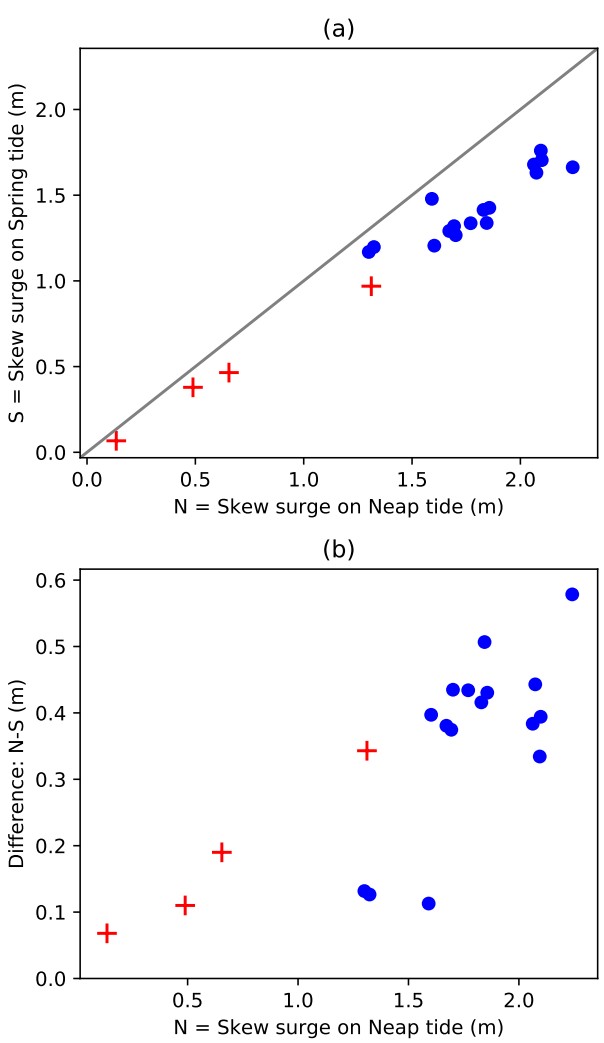

**Figure E1.** (a) Skew surge realised on spring tide ("S", Y-axis) vs. skew surge realised on neap tide ("N", X-axis) (b) Difference (N-S) vs. N. Blue dots show our experiments using atmospheric forcing from the 16 most extreme Sheerness events in the HadGEM3-GC3-MM simulation. Red crosses show data from Williams et al.(2016).

As discussed in the main text, such experiments have been conducted before by Williams et al. (2016); their results are shown by the red crosses in Fig. E1. Williams et al. (2016) also show scatter plots of observed skew surge and tide. In Fig. E2 we show our model results overlain on a reproduction of the Sheerness panel from their figure S2. It can be seen that our model results do not look out of context compared to the observations in terms of the negative correlation. For reference the extreme (entirely artificial) case of simulated tide-surge interaction is also shown, in which no tidal forcing is included (see §6.1). In reality, of course, the tide is always present.



Natural Hazards
and Earth System
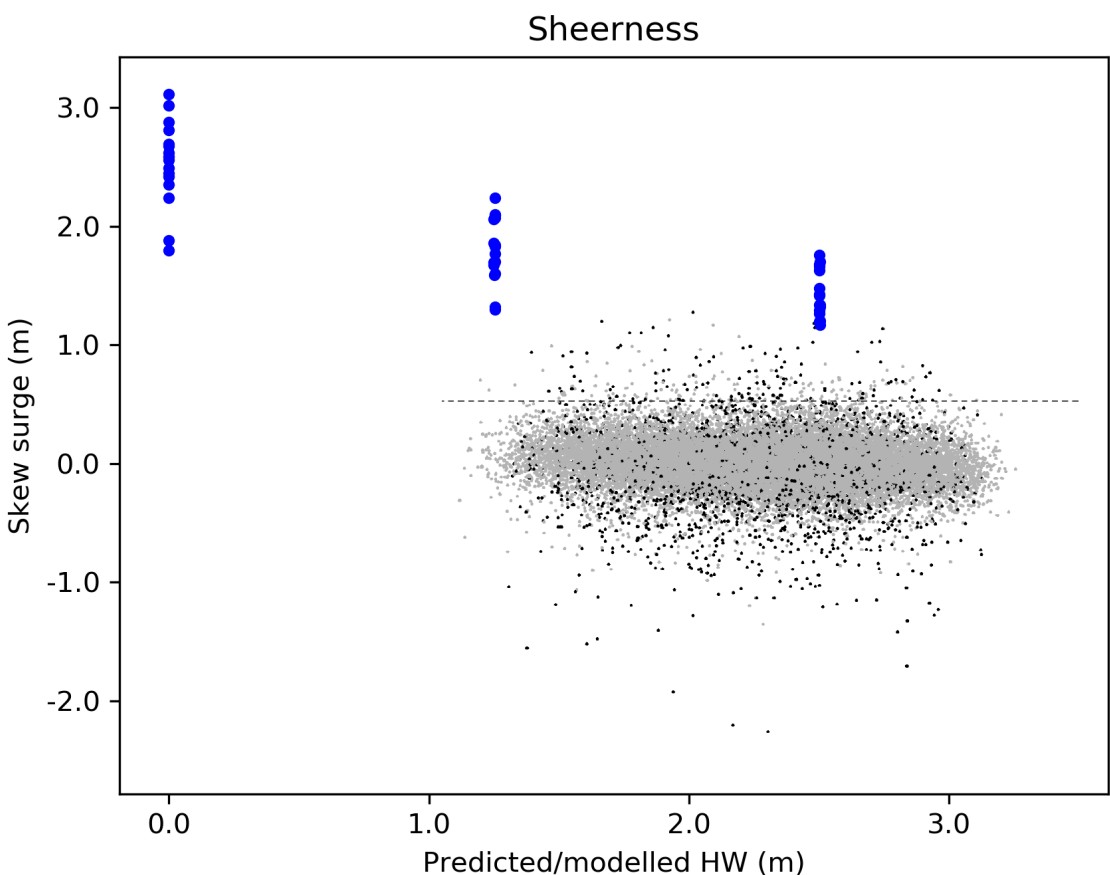

**Figure E2.** Our skew-surge/tide interaction results for Sheerness overlain on a reproduction of Williams et al.(2016), their figure S2. The results for a modelled high water of zero are included for comparison. They come from a very artificial simulation in which no tidal forcing is included. In reality, of course, the tide is always present.

*Author contributions.* TH devised the experiment, peformed most of the new analysis, and wrote the article. SDW performed the original CFB2018 analysis on the tide gauge data and replicated it for the simulation.

*Competing interests.* The authors declare no competing interests.



*Acknowledgements.* This work was supported by the Met Office Hadley Centre Climate Programme funded by BEIS and Defra. Special
thanks to Jenny Sansom for helpful discussions and for providing some of the data used in the evaluation. Thanks to Erik van Meijgaard
and Andreas Sterl at KNMI for comments on an early draft and for permission to use their RACMO-based atmospheric reconstruction of
the 1953 event, and to Mark Pickering at National Oceanography Centre Southampton for helping to supply the data. Thanks to Graham
Siggers (HR Wallingford) and Matt Palmer (Met Office Hadley Centre) for helpful comments on an early draft of this paper. Thanks to Jeff
Ridley for help with the HadGEM3-GC3-MM data. Thank you to Jon Tawn, Simon Brown and Rob Shooter for guidance with extreme value
modelling. This publication contains public sector information licensed under the Open Government Licence v3.0.



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
