# Peer review of "Towards using state-of-the-art climate models to help constrain estimates of unprecedented UK storm surges"

_Natural Hazards and Earth System Sciences, 2021_

## Referee Comment (RC1)

Review of

**Towards using state-of-the-art climate models to help constrain estimates of unprecedented UK storm surges**

by T. Howard and S. Williams

**Recommendation**

**Minor / major revisions** - dependign on editor's decision on investigating a Gumbel fit (see Discussion section below).

**Synopsis**

The paper investigates whether it is possible to exploit long climate model runs to estimate long return times of surges along the UK coast. To this end, the output of a 483-year integration of HadGEM3-GC3-MM is used to force the CS3 barotropic surge model. CS3 results are then compared to the (much shorter) observational records. Extreme Value Theory is used to infer long return times. A great deal of effort is put into estimation of the parameters of a GEV fitted to the model results. The shape parameter of a GEV distribution is found to be the source of the largest uncertainty in the return-level estimate.

The paper mainly deals with the skew surge because this measure has been shown to be independent of the tidal phase. In the case of Shearness, the modelling results contradict this view. The skew surge created by a specific wind storm differs by 20%, depending on whether the wind storm occurred during neap tide or during spring tide, respectively.

**Discussion**

Coastal protection works (dykes, flood barriers, etc) should protect against events of a magnitude that has not been reached during the observational record. Typically, such works are designed to protect against a water level occurring once in 1000 or 10,000 years. This time is much longer than the length of the observational records, which are usually not longer than 100 years. Extrapolating over orders of magnitude naturally leads to large uncertainties. Replacing observations by model-generated series, which often are much longer than the observational record, is therefore a good idea. The approach has been introduced nearly 20 years ago by Van Den Brink et al. (2004): Improving $10^4$-year surge level estimates using data of the ECMWF seasonal prediction system. Geophys. Res. Lett., 31, L17210, doi: 10.1029/2004GL020610. This fact should be acknowledged in the present paper.

The authors fit the three parameter (location, shape, scale) GEV distribution to the model results and find that the largest uncertainty comes from the scale parameter. That the scale parameter is the most uncertain parameter in a GEV fit is well known. It is most heavily determined by the highest values in the annual-maximum series. A way around is to use the two parameter (location and scale) Gumbel distribution. Especially for long time series, it often gives superiour results. Looking at Fig. 2c and d, a value of zero for the scale parameters is not inconsistent with at least the observational estimates. Whether the Gumbel distribution is a good approximation can be tested by the procedure explained in Van den Brink and Können (2011): Estimating 10000-year return values from short time series. Int. J. Climatol., 31:115-126, doi: 10.1002/joc.2047.

I am aware of the fact that it would require a lot of work to calculate the Gumbel-fits and test

for its suitability. However, I feel that the results shown in the paper could be much improved by eliminating the uncertainty that is inherent in the scale parameter. The editor should decide whether (s)he requires this analysis to be done.

**Detailed comments**

The paper is very well written, and I have only a few minor comments.

**p 12, l 300/301** Discussing Fig. 2c you refer to the "spread of the shape parameters diagnosed [...] from the simulation", but I cannot see any spread in the simulation-derived shape parameters. Spread is only depictied for the CFB estimates. Please clarify.

**sec. 6.1 / Fig. 6** You introduce a kernel. How did you obtain it? A short explanation of the procedure would be helpful. I am also not sure about the purpose of the kernel. Is it to extend the length of an episode, or its magnitude?

---

## Referee Comment (RC3)

**Discussion of**
**Towards using state-of-the-art climate models to help constrain estimates of unprecendented UK storm surges**

by Tom Howard and Simon David Paul Williams

**Discussants:** Eleanor D'Arcy and Jonathan Tawn

August 27, 2021

**1 Overview**

This paper proposes using climate model simulations to aid with still water level return level estimation and address problems that arise with the statistical approach when tide gauges have short record lengths. The model they present is the HadGEM3-GC3-MM to generate a dataset of 483-year present-day storm surges at sites on the UK tide gauge network. They compare the skew surge simulations to using only observations when fitting extreme value models by estimating parameters. The spatial distribution of the parameters for each dataset are generally well correlated. However, there is a negative bias in the simulation approach in the shape parameter estimate of the generalised Pareto distribution (GPD); the authors discuss this in detail and suspect it is due to a pitfall in the shelf sea model (CS3).

The paper also investigates the interaction between skew surge and peak tide at Sheerness. They study the effect that changes in timing between atmospheric forcings and tide has on skew surge. Additionally, they review the independence assumption of skew surge and peak tide used in the JPM. Using their model simulations, they show that extreme skew surges are more likely to occur on neap tide - this agrees with the results of Williams et al. (2016) (supplementary material).

**2 General Comments**

The paper is well written, in the proceeding sections we have discussed parts of the paper that pose interesting areas for future research and noted some technical corrections. The results are well presented and the figures well explained, giving a clear justification for the proposed model. The appendix provides strong support for ideas mentioned in the main paper. The authors have recognised potential downfalls with different aspects of the model and presented some initial investigation into these (for example, the discussion of why the shape parameter is more negative for simulations on pg. 15/16).

**3 Specific Comments**

**3.1 Comparison of Model and Observed Data**

Figure B1 shows some major departures between the model and observed data across the distribution of skew surges but particularly in the tails. In no sites is the model giving as high quantiles as the observed data. However, these departures have a systematic feature which is consistent over spatial regions, e.g., south-west and north-west UK. This suggests that it should be possible to account for these departures through a smooth spatial function which maps the differences in quantiles between the observations and model data. With this adjustment it is possible that the currently identified under-estimation may be corrected before making the tail based GEV/GPD comparisons you draw.

**3.2 Shape Parameters with Similar Spatial Patterns**

One of your exciting findings is that the spatial pattern of the shape parameter estimates is similar for the CFB2018 estimates from observed data and your estimates from the model, but with a systematic bias between them. This suggests using an alternative to the CFB2018 approach by explicitly exploiting this finding. Let $\xi_{obs}(x)$ and $\xi_{model}(x)$ be the shape parameters for the observed and model data respectively, for all gauged sites $x$. What you are saying is that you believe in the spatial variation of $\xi_{model}(x)$ but not its mean value. So you believe that $\xi_{obs}(x) = \xi_{model}(x) + \xi_{bias}$, where $\xi_{bias}$ is a fixed constant that does not vary over $x$. Your estimates indicate that $\xi_{bias} > 0$. Fixing estimates of $\xi_{model}(x)$ for all $x$ and fitting the function of $\xi_{obs}(x)$ over $x$ now means only one parameter, $\xi_{bias}$, needs estimating. This could lead to substantial reductions in the uncertainty of $\xi_{obs}(x)$ estimates over $x$.

**3.3 Penalised Likelihood**

The shape parameter estimates in Figure 2 (d) based on the 483 years of model data show some site-to-site variations which are more pronounced than the broader smooth variations across coastlines. This suggests that they would also benefit from the penalty-based approach used in CFB2018 work.

At a few points you state that the prior/penalty is subjective and that this is a disadvantage. Yet you also point that the smooth pattern of the shape parameter estimates this gives for the observed data agrees well with the similar unpenalised estimates using model data. You say this is a really positive feature of the model data, we would also take this as supporting the value of the process of creating penalised estimates from the observed data. The penalty is giving something meaningful, so the effect of the claimed "subjectivity" is positive for the observed data analysis. The prior that was selected in the CFB2018 work was not subjective in the traditional sense of a subjective prior in Bayesian methods. It was actually a data-based prior which corresponds to an empirical Bayesian prior, using all the information that separately estimated shape parameters for UK skew surge provide. The effect of this was simply to move shape parameter estimates more towards the UK average, with the larger changes coming for sites with shorter record lengths.

**3.4 Investigation of Interaction between Tide and Skew Surge**

We are really pleased to see the use of numerical models to explore systematically the widely claimed property that skew surges are independent of their associated peak tidal values. The work in Section 5 where you explore the timing effects of skew surge events relative to tides is very illuminating. It adds greatly to the existing empirical evidence for this feature at Sheerness. It would be interesting to have your opinions about what interactions are expected elsewhere in the UK given that the CFB2018 estimates over UK sites all assume independence. Since this analysis is based on simulations from HadGEM3-GC3-MM, this presents a physically-based justification for dependence between skew surge and peak tide at Sheerness. It suggests future research is required to investigate this. It may be that interaction occurs to some extent everywhere, but at a practical level it is not important apart from some locations.

In current work we have been investigating this feature empirically at a limited number of sites. Here we report some of the methods and findings using data from the tide gauges at Sheerness and Heysham (located close to Workington which you analyse). We study data from Heysham in 1964-2016, of which 17.5% is missing, and 1980-2016 at Sheerness, where 9.1% is missing.

We define extreme skew surges as exceedances of the 0.95 quantile at each site. Figure 1 shows scatter plots of extreme skew surges against their associated ranked peak tide; these are ranked so that 1 corresponds to the smallest observation and $n$ is the largest, where $n$ is the total number of observations. If the two components were independent, we expect extreme skew surges to be uniformly distributed over ranks. We test this using a Kolmogorov-Smirnov test for uniformity; the $p$ value at Heysham is 0.0224 whilst at Sheerness it is $1.40 \times 10^{-7}$. Clearly the $p$ value at Sheerness is much smaller, and provides statistical evidence that extreme skew surges are not independent of peak tide. However, at Heysham, there is not sufficient evidence to reject the claim of independence at the 0.01 significance level. This is clear in Figure 1, where more extreme skew surges occur on lower tides at Sheerness - this agrees with your findings and those of Williams et al. (2016).

We investigate this further by looking at skew surge and peak tide dependence on a month-by-month basis, using ideas from Williams et al. (2016). Figure 2 compares the distribution of peak tides associated

[Figure]

Figure 1: Extreme skew surge observations against ranked peak tide at Heysham (left) and Sheerness (right).

[Figure]

Figure 2: Monthly distributions of peak tides at Sheerness in February, May, August and October. The probability density function of all peak tides (black) and peak tides associated with extreme skew surge (red) are interpolated onto each distribution.

with all skew surges and the distribution of peak tides associated with extreme skew surges for February, May, August and October. If peak tide and skew surge are independent, these two distributions should be the same, up to sampling variation. We estimate the probability density function (pdf) using a Gaussian kernel density estimate, and use an Anderson-Darling test to check if peak tides come from the same distribution as peak tides associated with extreme skew surges. Figure 2 highlights how the dependence between skew surge and peak tide is changing with the time of year; the distribution of peak tides and the peak tides associated with extreme skew surge are most different in May and least different in February. In May, the mode of the distribution of peaks tides associated with extreme skew surges has shifted to a lower value than the distribution of all peak tides. Results from the Anderson Darling test for every month tell us there is insufficient evidence to reject the null hypothesis that peak tides and the peak tides associated with extreme skew surge come from the same distribution in February, March, September, November and December. In these months, we conclude it is reasonable to assume skew surge and peak tide are independent. In the remaining months, we find sufficient evidence to suggest the two components are dependent. We believe this poses a really interesting area for further research.

In D'Arcy et al. (2021) we fit a GPD (see Coles (2001) for details) to extreme skew surges that accounts for seasonal variations, since they are more extreme in the winter. Our results show this is an improvement on a standard GPD fit, where skew surges are assumed to be independent of peak tide. Here, we investigate adding a covariate of peak tide into our skew surge model to account for the dependence found at Sheerness. D'Arcy et al. (2021) defines extreme skew surges as exceedances of the monthly 0.95 quantile. Non-stationarity is accounted for in the scale parameter of the GPD through a daily covariate $d = 1, \ldots, 365$:

$$\sigma_d = a + b\sin\left(\frac{2\pi}{365}(d - \phi)\right) \tag{3.1}$$

for $a, b, \phi \in \mathbb{R}$ parameters to be estimated. Note that the shape parameter of the GPD is fixed across

months. To account for the dependence between skew surge and tide, we now also consider the following parameterisation on the scale parameter:

$$\sigma_{d,t} = a + b \sin\left(\frac{2\pi}{365}(d - \phi)\right) + ct \tag{3.2}$$

where $c \in \mathbb{R}$ is another parameter to be estimated, and $t$ the associated peak tide observation. We fit a GPD with both (3.1) and (3.2) formulations to extreme skew surges at Heysham and Sheerness. The Akaike information criteria (AIC), frequently used for model selection, suggests that formulation (3.1) gives a better model fit at Heysham, i.e., an independence conclusion is supported by this analysis. Whereas, AIC suggests that the parametrisation in equation (3.2) yields a better fit at Sheerness. By ordering peak tide observations from smallest to largest and calculating $\sigma_{d,t}$ at its winter peak ($d = 365$) for each value, we observe a 15% reduction in the scale parameter as tides increase at Sheerness, which corresponds to an equal percentage reduction in the skew surge quantiles for excesses of the December skew surge threshold. We also compare the model fits at each site using a Likelihood Ratio Test. At Heysham the $p$ value is 0.657 which provides insufficient evidence to reject the null hypothesis, that is the simpler model (in equation (3.1)) is sufficient. Whereas, at Sheerness we get a much smaller $p$ value of 0.059, which provides statistically significant evidence at the 0.1 significance level to reject the null hypothesis and conclude that the more complex model is required. This highlights the importance of accounting for skew surge and peak tide dependence at sites where the independence assumption is not justified.

**3.5 Ungauged Sites**

We feel you do not do full justice to your developments given the focus is on comparisons made at gauged sites with long and trustworthy records. The real value in modelled data is the ability to give estimates at other sites. This could be the focus of a natural follow up paper.

**3.6 Other Comments**

1. In the abstract, you say "results suggest an event of this magnitude has an expected frequency of about 1 in 500 years at [Sheerness]" when referring to the North Sea floods of 1953. In the paper, the only result to show this is presented in Section 6.2: "the fact that the 483-year surge-only simulation produces more than one event of comparable magnitude to the simulated 1953 event suggest the return period of the 1953 atmospheric forcing is less than 483 years." If this is the justification, it would be better to more formally quantify this using your statistical models.

2. In the Introduction (line 51) you list assumptions required to fit extreme value models as 'events are effectively random and statistically independent of each other.' But what about events being identically distributed (or stationary), it is clear that tide and storm surge (or skew surge) are not stationary as both process exhibit seasonality. Instead of "effectively random" it would be more mathematically correct to say "stochastic." Extreme value methods also handle dependence, so "independent of each other" is not formally required.

3. On line 84 climate change is discussed. Tide gauge observations will exhibit an approximately linear mean trend due to sea level rise, it is unclear whether this has been removed before comparison with the simulations in Section 4. It is likely that removing this change will not change the results significantly, but it is important to remove this non-stationary effect before fitting extreme value models.

4. Figure C1 shows the reduction in uncertainty on shape parameter when record lengths are increased in the tide gauge network, but it would be nice to see this for more record lengths that are equally spaced (say 10 to 500 years in increments of 10 years) to really highlight this - with measures of uncertainty as in Figure C1 (d).

**4 Technical Corrections**

1. Table 1 gives a list of useful acronyms and symbols, it would be help to include what 'GC3' and 'MM' stand for in 'HadGEM3-GC3-MM' as it is not mentioned in text.

2. Equation (2) has a surplus close bracket.

3. Equation (6), should this derivative be evaluated at $L = \log(u)$ rather than $y = u$ since $y$ is the return level (as in equation (1))?

4. Figures 2 (c) and (d) would benefit from having 95% confidence intervals on shape parameters using the model data.

5. Line 283: We think the wrong figure is referenced here.

6. Line 319: "They used . . . .time-series." This has nothing to do with how CFB2018 estimate the shape parameters and should be cut. It only links to the derivation of SWL return levels.

7. Line 349: You hint that the reason for the disagreement between the methods could be due the short observed data series. But Figure B1 shows major departures between the observed data and the model data in the body of the distributions to a level that could in no way be due to short observed series and must be that the model data does not reproduce well the observed data. Linked to this, in discussing Figure 1 you say the agreement at both sites is "excellent". With the amount of data at Sheerness it is clear that there are major disagreements that cannot be accounted by sampling variations.

8. Line 421: You suddenly mention a "kernel" to spread the duration of events but fail to provide any information. A little detail would be helpful here.

9. Figure E2: A description of the different points would be useful.

**References**

Coles, S. G. (2001). *An Introduction to Statistical Modeling of Extreme Values*. Springer, London.

D'Arcy, E., Tawn, J., Joly-Laugel, A., and Sifnioti, D. E. (2021). Accounting for seasonality in extreme sea level estimation *(in preparation)*.

Williams, J., Horsburgh, K. J., Williams, J. A., and Proctor, R. N. (2016). Tide and skew surge independence: New insights for flood risk. *Geophysical Research Letters*, 43(12):6410–6417.

---

## Author Comment (AC2)

[revised manuscript text omitted]

An obvious advantage of this approach is that the model is based on verifiable real-world physics. Many climate model simulations extend over periods longer than the tide-gauge record. In particular, in order to evaluate model performance, modellers use control simulations (with greenhouse gas forcing fixed at either pre-industrial or present-day levels) which may extend over many hundreds or even thousands of years. Ensemble simulations provide another potential source of data effectively covering a much longer period than the observations. Using the data from such simulations provides a further line of evidence in the effort to predict the magnitude and frequency of unprecedented events. Van den Brink et al. (2004) used this approach to simulate storm surges at Hoek van Holland using the ECMWF seasonal forecast ensemble, successfully reducing the uncertainty in the 10000-year return level by a factor of four compared to using the observations alone. 
[revised manuscript text omitted]
 den Brink, H., Können, G., Opsteegh, J., van Oldenborgh, G. J., and Burgers, G.: Improving $10^4$-year surge level estimates using data of the ECMWF seasonal prediction system, Geophysical Research Letters, 31, 2004.

van Meijgaard, E., van Ulft, L., van de Berg, W., Bosveld, F. C., van den Hurk, B., Lenderink, G., and Siebesma, A.: Technical report ; TR - 302. The KNMI regional atmospheric climate model RACMO version 2.1, [Available online at http://bibliotheek.knmi.nl/knmipubTR/TR302.pdf], cited 2020.

Wadey, M. P., Haigh, I., Nicholls, R. J., Brown, J. M., Horsburgh, K., Carroll, B., Gallop, S. L., Mason, T., Bradshaw, E., et al.: A comparison of the 31 January–1 February 1953 and 5–6 December 2013 coastal flood events around the UK, Frontiers in Marine Science, 2, 84, 2015.

Walters, D., Baran, A. J., Boutle, I., Brooks, M., Earnshaw, P., Edwards, J., Furtado, K., Hill, P., Lock, A., Manners, J., et al.: The Met Office Unified Model global atmosphere 7.0/7.1 and JULES global land 7.0 configurations, Geoscientific Model Development, 12, 1909–1963, 2019.

Weibull, W.: A statistical theory of strength of materials, IVB-Handl., 1939.

Williams, J., Horsburgh, K. J., Williams, J. A., and Proctor, R. N.: Tide and skew surge independence: New insights for flood risk, Geophysical Research Letters, 43, 6410–6417, 2016.

Williams, K., Copsey, D., Blockley, E., Bodas-Salcedo, A., Calvert, D., Comer, R., Davis, P., Graham, T., Hewitt, H., Hill, R., et al.: The Met Office global coupled model 3.0 and 3.1 (GC3. 0 and GC3. 1) configurations, Journal of Advances in Modeling Earth Systems, 10, 357–380, 2018.

Williams, K. D., Harris, C. M., Bodas-Salcedo, A., Camp, J., Comer, R. E., Copsey, D., Fereday, D., Graham, T., Hill, R., Hinton, T., Hyder, P., Ineson, S., Masato, G., Milton, S. F., Roberts, M. J., Rowell, D. P., Sanchez, C., Shelly, A., Sinha, B., Walters, D. N., West, A., Woollings, T., and Xavier, P. K.: The Met Office Global Coupled model 2.0 (GC2) configuration, Geoscientific Model Development, 8, 1509–1524, https://doi.org/10.5194/gmd-8-1509-2015, 2015.

Woollings, T., Hannachi, A., and Hoskins, B.: Variability of the North Atlantic eddy-driven jet stream, Quarterly Journal of the Royal Meteorological Society, 136, 856–868, 2010.

---

## Author Response (AR1)

All review comments and responses to

"Towards using state-of-the-art climate models to help constrain estimates of unprecedented UK storm surges"

By Tom Howard and Simon David Paul Williams

History of the review (taken from the NHESS website)

RC1:

**Review of**
**Towards using state-of-the-art climate models to help constrain estimates of unprecedented UK storm surges by T. Howard and S. Williams**

**Recommendation**

**Minor / major revisions** - dependign on editor's decision on investigating a Gumbel fit (see Discussion section below).

**Synopsis**

The paper investigates whether it is possible to exploit long climate model runs to estimate long return times of surges along the UK coast. To this end, the output of a 483-year integration of HadGEM3-GC3-MM is used to force the CS3 barotropic surge model. CS3 results are then compared to the (much shorter) observational records. Extreme Value Theory is used to infer long return times. A great deal of effort is put into estimation of the parameters of a GEV fitted to the model results. The shape parameter of a GEV distribution is found to be the source of the largest uncertainty in the return-level estimate.

The paper mainly deals with the skew surge because this measure has been shown to be independent of the tidal phase. In the case of Shearness, the modelling results contradict this view. The skew surge created by a specific wind storm differs by 20%, depending on whether the wind storm occurred during neap tide or during spring tide, respectively.

**Discussion**

Coastal protection works (dykes, flood barriers, etc) should protect against events of a magnitude that has not been reached during the observational record. Typically, such works are designed to protect against a water level occurring once in 1000 or 10,000 years. This time is much longer than the length of the observational records, which are usually not longer than 100 years. Extrapolating over orders of magnitude naturally leads to large uncertainties. Replacing observations by model-generated series, which often are much longer than the observational record, is therefore a good idea. The approach has been introduced nearly 20 years ago by Van Den Brink et al. (2004): Improving $10^4$-year surge level estimates using data of the ECMWF seasonal prediction system. Geophys. Res. Lett., 31, L17210, doi: 10.1029/2004GL020610. This fact should be acknowledged in the present paper.

The authors fit the three parameter (location, shape, scale) GEV distribution to the model results and find that the largest uncertainty comes from the scale parameter. That the scale parameter is the most uncertain parameter in a GEV fit is well known. It is most heavily determined by the highest values in the annual-maximum series. A way around is to use the two parameter (location and scale) Gumbel distribution. Especially for long time series, it often gives superiour results. Looking at Fig. 2c and d, a value of zero for the scale parameters is not inconsistent with at least the observational estimates. Whether the Gumbel distribution is a good approximation can be tested by the procedure explained in Van den Brink and K¨onnen (2011): Estimating 10000-year return values from short time series. Int. J. Climatol., 31:115-126, doi: 10.1002/joc.2047.

I am aware of the fact that it would require a lot of work to calculate the Gumbel-fits and test for its suitability. However, I feel that the results shown in the paper could be much improved by eliminating the uncertainty that is inherent in the scale parameter. The editor should decide whether (s)he requires this analysis to be done.

**Detailed comments**

The paper is very well written, and I have only a few minor comments.

**p 12, l 300/301** Discussing Fig. 2c you refer to the "spread of the shape parameters diagnosed [...] from the simulation", but I cannot see any spread in the simulation-derived shape parameters. Spread is only depictied for the CFB estimates. Please clarify.

**sec. 6.1 / Fig. 6** You introduce a kernel. How did you obtain it? A short explanation of the procedure would be helpful. I am also not sure about the purpose of the kernel. Is it to extend the length of an episode, or its magnitude?
* * *
RC2:

I obviously confused 'shape' and 'scale' in my comment. In the second para of the Discussion section it is the scale paramter that causes the largest uncertainty and that seems to be compatble with zero. Sorry for any confusion that may have arisen..

AC1:

Response to review by Andreas Sterl of *Towards using state-of-the-art climate models to help constrain estimates of unprecedented UK storm surges.*

*by T. Howard and S. Williams, submitted 2021*

*(henceforth HW21)*

This response: Tom Howard Aug 2021

Thank you for your review and for raising a very interesting question. I have amended the draft to address the minor points, and included a discussion of your main point further below. I will attempt to upload the amended draft to accompany this response.

Regarding the minor points first (quotes from the review are shown in red).

The approach has been introduced nearly 20 years ago by Van Den Brink et al. (2004): Improving 10^4 year surge level estimates using data of the ECMWF seasonal prediction system. Geophys. Res. Lett., 31, L17210, doi: 10.1029/2004GL020610. This fact should be acknowledged in the present paper.

I apologise for failing to cite this very relevant paper in the first draft. I have now acknowledged this contribution.

Discussing Fig. 2c you refer to the "spread of the shape parameters diagnosed

[...] from the simulation", but I cannot see any spread in the simulation-derived shape

parameters. Spread is only depicted for the CFB estimates. Please clarify.

Thank you for pointing this out. I was referring to the spread associated with the spatial variations, rather than the uncertainty. I have clarified this in the paper as follows:

"The spread (i.e. the size of the spatial variations) of the shape parameters…"

You introduce a kernel. How did you obtain it? A short explanation of the

procedure would be helpful. I am also not sure about the purpose of the kernel. Is it to

I have added the following sentences explaining the choice and purpose of the kernel:

"The kernel was designed to represent the important features of the RACMO-driven surge-only simulation, i.e., the approximate duration and shape of the time series plot. The purpose of convolution with the kernel is to identify those events which correlate well (in terms of their time series plot) with the RACMO-driven simulation, in other words, events which not only produce a large surge, but are also of comparable duration to the RACMO-driven simulation. The kernel was not used to modify events, but simply to identify significant ones."
* * *
Turning now to your main point regarding whether to fix the shape parameter at zero. This is prompted by the two papers by van den Brink and Können:

van den Brink and Können (2011): Estimating 10000-year return values from short time series. Int. J. Climatol., 31:115-126, doi: 10.1002/joc.2047 , and their related 2008 paper (both referred to here as vdB&K).

Thank you for reminding me of vdB&K's very interesting approach. I spent a long time (back in 2017) studying their 2008 paper, but it is only now, in testing the method on our own data in order to complete this response, that I am beginning to understand it.

HW21 was rooted in the methodology of CFB2018, which forms the current UK guidance on sea level extremes:

https://assets.publishing.service.gov.uk/media/603652cce90e0740b7caac9d/Coastal_flood_boundary_conditions_for_the_UK_2018_update_-_technical_report.pdf

(full reference in HW21). CFB2018 was developed in consultation with Professor Jonathan Tawn:

https://www.maths.lancs.ac.uk/~tawn/

Jonathan advised me not to fix the shape parameter as this gives false confidence intervals. The position is laid out in the textbook by Stuart Coles (full reference in HW21). Coles provides an illustration of a case similar to the case in section 4.5 of vdB&K (2011), and Coles discusses it as follows (Coles page 64)<Quote>:

**Reduction of uncertainty is desirable, so that if the Gumbel model could be trusted its inferences would be preferred. But can the model be trusted? The extremal types theorem provides support for modelling block maxima with the GEV family of which the Gumbel family is a subset. The data** [in Coles's example 3.4.1, which, like vdB&K(2011) example 4.5, has an estimated shape parameter close to zero] **suggest that a Gumbel model is plausible, but this does not imply that other models are not. Indeed, the maximum likelihood estimate within the GEV family is not in the Gumbel family (although, in the sense that the estimated shape parameter is close to zero, it is "close").** **There is no common agreement about this issue,** **but the safest option is to accept there is uncertainty about the value of the shape parameter --- and hence whether the Gumbel model is correct or not --- and to prefer the inference based on the GEV model.** **The larger measures of uncertainty generated by the GEV model then provide a more realistic quantification of genuine uncertainties involved in model extrapolation.**

<End Quote> (The red highlighting is mine)

On the other hand, we know that unconstrained GEV fits to short record lengths can give implausible shape parameters. One way to fix this is to put a prior on the shape parameter (see for example Martins and Stedinger 2000, full reference in HW21). This was the approach used in CFB2018 (again under advice from Jonathan Tawn).

In HW21 we find that the spatial variations in the shape parameter diagnosed from the tide gauges are also seen in the shape parameter as diagnosed from the simulation. We have argued that this finding supports the credibility of the spatial variations. Thus, it would be inconsistent to adopt an approach of fixing the shape parameter at zero within HW21, but for completeness I have tested a Gumbel fit to the annual maxima following the method of vdB&K and I show the results here.

I have used our data to make plots analogous to vdB&K 2008 Figure 3, i.e., their $\Delta \hat{X}_n$ vs. Gumbel variate plot, comparing Gumbel fits to the annual maxima with the fits used in our paper and in CFB2018. Fig. 1 is the plot for the observations:

[Figure]

Fig. 1. Each data point represents a UK tide gauge. The points labelled "CFB" show $\Delta\hat{X}_n$ where $\hat{F}$ is determined by the CFB2018 method (constrained GPD fit to peaks over a threshold, as described in HW21), and the points labelled "Gumbel" show $\Delta\hat{X}_n$ where $\hat{F}$ is determined by a Gumbel fit to the annual maxima.

I can't see strong evidence here that the Gumbel fit is better overall. I was a bit concerned that I had not been rigorous about checking for independence of the events shown. This would not be straightforward to do with my current software setup, but a simple first fix is to miss out closely-neighbouring tide gauges. The following plots show results from every second tide gauge (Separation=2), every third tide gauge (Separation=3), etc. (Separation=1 is just a duplicate of the above plot).

[Figure]

Fig. 2. See main text.

Again, there does not appear to be much support for preferring the Gumbel fit.

(continued…)

I followed the same procedure for the simulated skew surges to make Figs 3 and 4.

[Figure]

Fig. 3. As Fig. 1, but for the simulated skew surges. Each data point represents a UK tide gauge. The points labelled "GEV" show $\Delta \hat{X}_n$ where $\hat{F}$ is determined by a GEV fit to the annual maxima, and the points labelled "Gumbel" show $\Delta \hat{X}_n$ where $\hat{F}$ is determined by a Gumbel fit to the annual maxima.

[Figure]

Fig. 4. As Fig. 2, but for the simulated skew surges.

Again, this does not seem to me to give support for preferring the Gumbel fit. Even if it did, I would not think it defensible to fix the shape parameter at a single value (e.g., zero). Looked at from the point of view of applying a prior to the shape parameter, fixing it at a single value seems to be equivalent to asserting that we are sure that no other value is plausible, and we are sure that the shape parameter does not vary by location. I cannot support either of those assertions.

Note to self: internal reference for figures: fig_KK

The reviewer responded with a personal communication which I have copied to the editor.

AC2: This was a diff.pdf between the original submission and the first revision. There seems little point in including it here because we later have the diff between the original submission and the second revision.

RC3:

Discussion of

**Towards using state-of-the-art climate models to help constrain estimates of unprecedented UK storm surges**

by Tom Howard and Simon David Paul Williams

Discussants: Eleanor D'Arcy and Jonathan Tawn

August 27, 2021

**Overview**

This paper proposes using climate model simulations to aid with still water level return level estimation and address problems that arise with the statistical approach when tide gauges have short record lengths. The model they present is the HadGEM3-GC3-MM to generate a dataset of 483-year present-day storm surges at sites on the UK tide gauge network. They compare the skew surge simulations to using only observations when fitting extreme value models by estimating parameters. The spatial distribution of the parameters for each dataset are generally well correlated. However, there is a negative bias in the simulation approach in the shape parameter estimate of the generalised Pareto distribution (GPD); the authors discuss this in detail and suspect it is due to a pitfall in the shelf sea model (CS3).

The paper also investigates the interaction between skew surge and peak tide at Sheerness. They study the effect that changes in timing between atmospheric forcings and tide has on skew surge. Additionally, they review the independence assumption of skew surge and peak tide used in the JPM. Using their model simulations, they show that extreme skew surges are more likely to occur on neap tide - this agrees with the results of Williams et al. (2016) (supplementary material).

**General Comments**

The paper is well written, in the proceeding sections we have discussed parts of the paper that pose interesting areas for future research and noted some technical corrections. The results are well presented and the figures well explained, giving a clear justification for the proposed model. The appendix provides strong support for ideas mentioned in the main paper. The authors have recognised potential downfalls with different aspects of the model and presented some initial investigation into these (for example, the discussion of why the shape parameter is more negative for simulations on pg. 15/16).

**Specific Comments**

**Comparison of Model and Observed Data**
Figure B1 shows some major departures between the model and observed data across the distribution of skew surges but particularly in the tails. In no sites is the model giving as high quantiles as the observed data. However, these departures have a systematic feature which is consistent over spatial regions, e.g., south-west and north-west UK. This suggests that it should be possible to account for these departures through a smooth spatial function which maps the

differences in quantiles between the observations and model data. With this adjustment it is possible that the currently identified under-estimation may be corrected before making the tail based GEV/GPD comparisons you draw.

**Shape Parameters with Similar Spatial Patterns**

One of your exciting findings is that the spatial pattern of the shape parameter estimates is similar for the CFB2018 estimates from observed data and your estimates from the model, but with a systematic bias between them. This suggests using an alternative to the CFB2018 approach by explicitly exploiting this finding. Let $\xi_{obs}(x)$ and $\xi_{model}(x)$ be the shape parameters for the observed and model data respectively, for all gauged sites $x$. What you are saying is that you believe in the spatial variation of $\xi_{model}(x)$ but not its mean value. So you believe that $\xi_{obs}(x) = \xi_{model}(x) + \xi_{bias}$, where $\xi_{bias}$ is a fixed constant that does not vary over $x$. Your estimates indicate that $\xi_{bias} > 0$. Fixing estimates of $\xi_{model}(x)$ for all $x$ and fitting the function of $\xi_{obs}(x)$ over $x$ now means only one parameter, $\xi_{bias}$, needs estimating. This could lead to substantial reductions in the uncertainty of $\xi_{obs}(x)$ estimates over $x$.

**Penalised Likelihood**

The shape parameter estimates in Figure 2 (d) based on the 483 years of model data show some site-to-site variations which are more pronounced than the broader smooth variations across coastlines. This suggests that they would also benefit from the penalty-based approach used in CFB2018 work.

At a few points you state that the prior/penalty is subjective and that this is a disadvantage. Yet you also point that the smooth pattern of the shape parameter estimates this gives for the observed data agrees well with the similar unpenalised estimates using model data. You say this is a really positive feature of the model data, we would also take this as supporting the value of the process of creating penalised estimates from the observed data. The penalty is giving something meaningful, so the effect of the claimed "subjectivity" is positive for the observed data analysis. The prior that was selected in the CFB2018 work was not subjective in the traditional sense of a subjective prior in Bayesian methods. It was actually a data-based prior which corresponds to an empirical Bayesian prior, using all the information that separately estimated shape parameters for UK skew surge provide. The effect of this was simply to move shape parameter estimates more towards the UK average, with the larger changes coming for sites with shorter record lengths.

**Investigation of Interaction between Tide and Skew Surge**

We are really pleased to see the use of numerical models to explore systematically the widely claimed property that skew surges are independent of their associated peak tidal values. The work in Section 5 where you explore the timing effects of skew surge events relative to tides is very illuminating. It adds greatly to the existing empirical evidence for this feature at Sheerness. It would be interesting to have your opinions about what interactions are expected elsewhere in the UK given that the CFB2018 estimates over UK sites all assume independence. Since this analysis is based on simulations from HadGEM3-GC3-MM, this presents a physically-based justification for dependence between skew surge and peak tide at Sheerness. It suggests future research is required to investigate this. It may be that interaction occurs to some extent everywhere, but at a practical level it is not important apart from some locations.

In current work we have been investigating this feature empirically at a limited number of sites. Here we report some of the methods and findings using data from the tide gauges at Sheerness and

Heysham (located close to Workington which you analyse). We study data from Heysham in 1964-2016, of which 17.5% is missing, and 1980-2016 at Sheerness, where 9.1% is missing.

We define extreme skew surges as exceedances of the 0.95 quantile at each site. Figure 1 shows scatter plots of extreme skew surges against their associated ranked peak tide; these are ranked so that 1 corresponds to the smallest observation and $n$ is the largest, where $n$ is the total number of observations. If the two components were independent, we expect extreme skew surges to be uniformly distributed over ranks. We test this using a Kolmogorov-Smirnov test for uniformity; the $p$ value at Heysham is 0.0224 whilst at Sheerness it is $1.40 \times 10^{-7}$. Clearly the $p$ value at Sheerness is much smaller, and provides statistical evidence that extreme skew surges are not independent of peak tide. However, at Heysham, there is not sufficient evidence to reject the claim of independence at the 0.01 significance level. This is clear in Figure 1, where more extreme skew surges occur on lower tides at Sheerness - this agrees with your findings and those of Williams et al. (2016).

We investigate this further by looking at skew surge and peak tide dependence on a month-by-month basis, using ideas from Williams et al. (2016). Figure 2 compares the distribution of peak tides associated

[Figure]

Figure 1: Extreme skew surge observations against ranked peak tide at Heysham (left) and Sheerness (right).

[Figure]

Figure 2: Monthly distributions of peak tides at Sheerness in February, May, August and October. The probability density function of all peak tides (black) and peak tides associated with extreme skew surge (red) are interpolated onto each distribution.

with all skew surges and the distribution of peak tides associated with extreme skew surges for February, May, August and October. If peak tide and skew surge are independent, these two distributions should be the same, up to sampling variation. We estimate the probability density function (pdf) using a Gaussian kernel density estimate, and use an Anderson-Darling test to check if peak tides come from the same distribution as peak tides associated with extreme skew surges. Figure 2 highlights how the dependence between skew surge and peak tide is changing with the time of year; the distribution of peak tides and the peak tides associated with extreme skew surge are

most different in May and least different in February. In May, the mode of the distribution of peaks tides associated with extreme skew surges has shifted to a lower value than the distribution of all peak tides. Results from the Anderson Darling test for every month tell us there is insufficient evidence to reject the null hypothesis that peak tides and the peak tides associated with extreme skew surge come from the same distribution in February, March, September, November and December. In these months, we conclude it is reasonable to assume skew surge and peak tide are independent. In the remaining months, we find sufficient evidence to suggest the two components are dependent. We believe this poses a really interesting area for further research.

In D'Arcy et al. (2021) we fit a GPD (see Coles (2001) for details) to extreme skew surges that accounts for seasonal variations, since they are more extreme in the winter. Our results show this is an improvement on a standard GPD fit, where skew surges are assumed to be independent of peak tide. Here, we investigate adding a covariate of peak tide into our skew surge model to account for the dependence found at Sheerness. D'Arcy et al. (2021) defines extreme skew surges as exceedances of the monthly 0.95 quantile. Non-stationarity is accounted for in the scale parameter of the GPD through a daily covariate $d = 1,\dots,365$:

$$\sigma_d = a + b\sin\left(\frac{2\pi}{365}(d - \phi)\right)$$

(3.1)

for $a, b, \varphi \in \mathbb{R}$ parameters to be estimated. Note that the shape parameter of the GPD is fixed across months. To account for the dependence between skew surge and tide, we now also consider the following parameterisation on the scale parameter:

$$\sigma_{d,t} = a + b\sin\left(\frac{2\pi}{365}(d - \phi)\right) + ct$$

(3.2)

where $c \in \mathbb{R}$ is another parameter to be estimated, and $t$ the associated peak tide observation. We fit a GPD with both (3.1) and (3.2) formulations to extreme skew surges at Heysham and Sheerness. The Akaike information criteria (AIC), frequently used for model selection, suggests that formulation (3.1) gives a better model fit at Heysham, i.e., an independence conclusion is supported by this analysis. Whereas, AIC suggests that the parametrisation in equation (3.2) yields a better fit at Sheerness. By ordering peak tide observations from smallest to largest and calculating $\sigma_{d,t}$ at its winter peak ($d = 365$) for each value, we observe a 15% reduction in the scale parameter as tides increase at Sheerness, which corresponds to an equal percentage reduction in the skew surge quantiles for excesses of the December skew surge threshold. We also compare the model fits at each site using a Likelihood Ratio Test. At Heysham the $p$ value is 0.657 which provides insufficient evidence to reject the null hypothesis, that is the simpler model (in equation (3.1)) is sufficient. Whereas, at Sheerness we get a much smaller $p$ value of 0.059, which provides statistically significant evidence at the 0.1 significance level to reject the null hypothesis and conclude that the more complex model is required. This highlights the importance of accounting for skew surge and peak tide dependence at sites where the independence assumption is not justified.

**Ungauged Sites**
We feel you do not do full justice to your developments given the focus is on comparisons made at gauged sites with long and trustworthy records. The real value in modelled data is the ability to give estimates at other sites. This could be the focus of a natural follow up paper.

**Other Comments**

1. In the abstract, you say "results suggest an event of this magnitude has an expected frequency of about 1 in 500 years at [Sheerness]" when referring to the North Sea floods of 1953. In the paper, the only result to show this is presented in Section 6.2: "the fact that the 483-year surge-only simulation produces more than one event of comparable magnitude to the simulated 1953 event suggest the return period of the 1953 atmospheric forcing is less than 483 years." If this is the justification, it would be better to more formally quantify this using your statistical models.

2. In the Introduction (line 51) you list assumptions required to fit extreme value models as 'events are effectively random and statistically independent of each other.' But what about events being identically distributed (or stationary), it is clear that tide and storm surge (or skew surge) are not stationary as both process exhibit seasonality. Instead of "effectively random" it would be more mathematically correct to say "stochastic." Extreme value methods also handle dependence, so "independent of each other" is not formally required.

3. On line 84 climate change is discussed. Tide gauge observations will exhibit an approximately linear mean trend due to sea level rise, it is unclear whether this has been removed before comparison with the simulations in Section 4. It is likely that removing this change will not change the results significantly, but it is important to remove this non-stationary effect before fitting extreme value models.

4. Figure C1 shows the reduction in uncertainty on shape parameter when record lengths are increased in the tide gauge network, but it would be nice to see this for more record lengths that are equally spaced (say 10 to 500 years in increments of 10 years) to really highlight this - with measures of uncertainty as in Figure C1 (d).

**Technical Corrections**

1. Table 1 gives a list of useful acronyms and symbols, it would be help to include what 'GC3' and 'MM' stand for in 'HadGEM3-GC3-MM' as it is not mentioned in text.

2. Equation (2) has a surplus close bracket.

3. Equation (6), should this derivative be evaluated at $L = \log(u)$ rather than $y = u$ since $y$ is the return level (as in equation (1))?

4. Figures 2 (c) and (d) would benefit from having 95% confidence intervals on shape parameters using the model data.

5. Line 283: We think the wrong figure is referenced here.

6. Line 319: "They used ....time-series." This has nothing to do with how CFB2018 estimate the shape parameters and should be cut. It only links to the derivation of SWL return levels.

7. Line 349: You hint that the reason for the disagreement between the methods could be due the short observed data series. But Figure B1 shows major departures between the observed data and the model data in the body of the distributions to a level that could in no way be due to short observed series and must be that the model data does not reproduce well the observed data. Linked to this, in discussing Figure 1 you say the agreement at both sites is "excellent". With the amount of data at Sheerness it is clear that there are major disagreements that cannot be accounted by sampling variations.

8. Line 421: You suddenly mention a "kernel" to spread the duration of events but fail to provide any information. A little detail would be helpful here.

9. Figure E2: A description of the different points would be useful.

**References**

Coles, S. G. (2001). *An Introduction to Statistical Modeling of Extreme Values*. Springer, London.

D'Arcy, E., Tawn, J., Joly-Laugel, A., and Sifnioti, D. E. (2021). Accounting for seasonality in extreme sea level estimation *(in preparation)*.

Williams, J., Horsburgh, K. J., Williams, J. A., and Proctor, R. N. (2016). Tide and skew surge independence: New insights for flood risk. *Geophysical Research Letters*, 43(12):6410–6417.
* * *
AC3:

Tom Howard    9 Sep. 21
**Author response to:**

**Discussion of**
**Towards using state-of-the-art climate models to help constrain estimates of unprecedented UK storm surges**

by Tom Howard and Simon David Paul Williams

Discussants: Eleanor D'Arcy and Jonathan Tawn

August 27, 2021

Thank you both very much for your thorough and interesting review. I have pasted in the whole discussion in dark orange font colour; my responses are shown in blue.

**Overview**

This paper proposes using climate model simulations to aid with still water level return level estimation and address problems that arise with the statistical approach when tide gauges have short record lengths. The model they present is the HadGEM3-GC3-MM to generate a dataset of 483-year present-day storm surges at sites on the UK tide gauge network. They compare the skew surge simulations to using only observations when fitting extreme value models by estimating parameters. The spatial distribution of the parameters for each dataset are generally well correlated. However, there is a negative bias in the simulation approach in the shape parameter estimate of the generalised Pareto distribution (GPD); the authors discuss this in detail and suspect it is due to a pitfall in the shelf sea model (CS3).

The paper also investigates the interaction between skew surge and peak tide at Sheerness. They study the effect that changes in timing between atmospheric forcings and tide has on skew surge. Additionally, they review the independence assumption of skew surge and peak tide used in the JPM. Using their model simulations, they

show that extreme skew surges are more likely to occur on neap tide - this agrees with the results of Williams et al. (2016) (supplementary material).

**General Comments**

The paper is well written, in the proceeding sections we have discussed parts of the paper that pose interesting areas for future research and noted some technical corrections. The results are well presented and the figures well explained, giving a clear justification for the proposed model. The appendix provides strong support for ideas mentioned in the main paper. The authors have recognised potential downfalls with different aspects of the model and presented some initial investigation into these (for example, the discussion of why the shape parameter is more negative for simulations on pg. 15/16).

**Specific Comments**

**Comparison of Model and Observed Data**

Figure B1 shows some major departures between the model and observed data across the distribution of skew surges but particularly in the tails. In no sites is the model giving as high quantiles as the observed data. However, these departures have a systematic feature which is consistent over spatial regions, e.g., south-west and north-west UK. This suggests that it should be possible to account for these departures through a smooth spatial function which maps the differences in quantiles between the observations and model data. With this adjustment it is possible that the currently identified under-estimation may be corrected before making the tail based GEV/GPD comparisons you draw.

Have added the following to the description accompanying Figure B1:

"Figure B1 shows some major departures between the model and observed data across the distribution of skew surges, but particularly in the tails. The model does not give higher quantiles than the observed data at any site."

And added the following to the main text:

"The model does not give higher quantiles than the observed data at any site."

Have added a section "Suggestions for further work", paraphrasing your comment in the following text:

"The model/observation departures seen in Fig. B1 have a systematic feature which is consistent over spatial regions, e.g., south-west and north-west UK. This suggests that it should be possible to account for these departures through a smooth spatial function which maps the differences in quantiles between the observations and model data. With this adjustment it is possible that the currently identified under-estimation may be corrected before making the tail-based GEV/GPD comparisons shown here."

On first reading your suggestion, I thought: "that would only correct the location and scale parameters, which we are suggesting be taken from the observations anyway --- it wouldn't make sense to correct the model shape parameters using the shorter observational records and then argue that the observational shape parameters should be replaced with those of the model". But then I remembered about the scale-shape compensation which can occur at the fitting stage. Is that the reason for your suggestion?

**Shape Parameters with Similar Spatial Patterns**

One of your exciting findings is that the spatial pattern of the shape parameter estimates is similar for the CFB2018 estimates from observed data and your estimates from the model, but with a systematic bias between them. This suggests using an alternative to the CFB2018 approach by explicitly exploiting this finding. Let $\xi_{obs}(x)$ and $\xi_{model}(x)$ be the shape parameters for the observed and model data respectively, for all gauged sites $x$. What you are saying is that you believe in the spatial variation of $\xi_{model}(x)$ but not its mean value. So you believe that $\xi_{obs}(x) = \xi_{model}(x) + \xi_{bias}$, where $\xi_{bias}$ is a fixed constant that does not vary over $x$. Your estimates indicate that $\xi_{bias} > 0$. Fixing estimates of $\xi_{model}(x)$ for all $x$ and fitting the function of $\xi_{obs}(x)$ over $x$ now means only one parameter, $\xi_{bias}$, needs estimating. This could lead to substantial reductions in the uncertainty of $\xi_{obs}(x)$ estimates over $x$.

I like that idea a lot! Have incorporated it in the main text. Hope that is OK. Have acknowledged your help in the acknowledgements section.

Line 344: have removed the phrase "without the subjectivity of a prior" and added the following:

"Given the need for some kind of constraint on the shape parameter when fitting observational records, use of shape parameters from a long simulation holds the promise of reducing uncertainties. For example, if we assume that the model-diagnosed spatial pattern of shape parameters is correct but uniformly biased by a scalar $\xi_{bias}$ (which does not vary over sites), $ \xi_{true}(x) = \xi_{model}(x) + \xi_{bias} $, where $x$ is a vector of sites, then we can use the observations from \emph{all sites} to estimate the one scalar parameter $\xi_{bias}$. This could lead to substantial reductions in the uncertainty of $\xi_{true}$ estimates over $x$."

**Penalised Likelihood**

The shape parameter estimates in Figure 2 (d) based on the 483 years of model data show some site-to-site variations which are more pronounced than the broader smooth variations across coastlines. This suggests that they would also benefit from the penalty-based approach used in CFB2018 work.

Thanks, have paraphrased this in the new "Suggestions for further work" section.

At a few points you state that the prior/penalty is subjective and that this is a disadvantage. Yet you also point that the smooth pattern of the shape parameter estimates this gives for the observed data agrees well with the similar unpenalised estimates using model data. You say this is a really positive feature of the model data, we would also take this as supporting the value of the process of creating penalised estimates from the observed data. The penalty is giving something meaningful, so the effect of the claimed "subjectivity" is positive for the observed data analysis.

I completely agree. I'm sorry if this did not come across clearly in the draft. Indeed, in some other experiments (not discussed in the paper) I tried unconstrained GEV fits to the annual maxima from the observations. This spoiled the strong correlation with the model-diagnosed shape parameters, so yes, completely agree that the CFB2018 penalisation process adds value over an unconstrained fit.

In the draft, we have the following text:

This strong correlation between the two spatial patterns of shape parameter diagnosed from independent sources (i.e. our model simulation and the tide-gauge data) is remarkable. It both supports the spatial pattern of the shape parameter as a real, physically-determined phenomenon (as opposed to a statistical artefact), **and gives further credibility to both the CFB2018 approach** and our model. [bold font is not in the draft].

The prior that was selected in the CFB2018 work was not subjective in the traditional sense of a subjective prior in Bayesian methods. It was actually a data-based prior which corresponds to an empirical Bayesian prior, using all the information that separately estimated shape parameters for UK skew surge provide. The effect of this was simply to move shape parameter estimates more towards the UK average, with the larger changes coming for sites with shorter record lengths.

Thank you for your guidance. I didn't previously understand that difference in the definition of subjective vs empirical. Have added your description to the draft as follows, and removed all references to "subjectivity"

"CFB2018 employed a data-based prior using all the information that separately estimated shape parameters for UK skew surge provide. The effect of this was simply to move shape parameter estimates more towards the UK average, with the larger changes coming for sites with shorter record lengths."

**Investigation of Interaction between Tide and Skew Surge**

We are really pleased to see the use of numerical models to explore systematically the widely claimed property that skew surges are independent of their associated peak tidal values. The work in Section 5 where you explore the timing effects of skew surge events relative to tides is very illuminating. It adds greatly to the existing empirical evidence for this feature at Sheerness. It would be interesting to have your opinions about what interactions are expected elsewhere in the UK given that the CFB2018 estimates over UK sites all assume independence. Since this analysis is based on simulations from HadGEM3-GC3-MM, this presents a physically-based justification for dependence between skew surge and peak tide at Sheerness. It suggests future research is required to investigate this. It may be that interaction occurs to some extent everywhere, but at a practical level it is not important apart from some locations.

Agreed. Have added a sentence to the "Suggestions for further work" section. For what it's worth, my guess is that interaction is stronger at Sheerness than elsewhere, and is associated with the surge and tide travelling a long way together (all the way down the east coast) in shallow water. But that is just a guess.

In current work we have been investigating this feature empirically at a limited number of sites. Here we report some of the methods and findings using data from the tide gauges at Sheerness and Heysham (located close to Workington which you analyse). We study data from Heysham in 1964-2016, of which 17.5% is missing, and 1980-2016 at Sheerness, where 9.1% is missing.

We define extreme skew surges as exceedances of the 0.95 quantile at each site. Figure 1 shows scatter plots of extreme skew surges against their associated ranked peak tide; these are ranked so that 1 corresponds to the smallest observation and $n$ is the largest, where $n$ is the total number of observations. If the two components were independent, we expect extreme skew surges to be uniformly distributed over ranks. We test this using a Kolmogorov-Smirnov test for uniformity; the $p$ value at Heysham is 0.0224 whilst at Sheerness it is $1.40 \times 10^{-7}$. Clearly the $p$ value at Sheerness is much smaller, and provides statistical evidence that extreme skew surges are not independent of peak tide. However, at Heysham, there is not sufficient evidence to reject the claim of independence at the 0.01 significance level. This is clear in Figure 1, where more extreme skew surges occur on lower tides at Sheerness - this agrees with your findings and those of Williams et al. (2016).

We investigate this further by looking at skew surge and peak tide dependence on a month-by-month basis, using ideas from Williams et al. (2016). Figure 2 compares the distribution of peak tides associated

[Figure]

Figure 1: Extreme skew surge observations against ranked peak tide at Heysham (left) and Sheerness (right).

[Figure]

Figure 2: Monthly distributions of peak tides at Sheerness in February, May, August and October. The probability density function of all peak tides (black) and peak tides associated with extreme skew surge (red) are interpolated onto each distribution.

with all skew surges and the distribution of peak tides associated with extreme skew surges for February, May, August and October. If peak tide and skew surge are independent, these two distributions should be the same, up to sampling variation. We estimate the probability density function (pdf) using a Gaussian kernel density estimate, and use an Anderson-Darling test to check if peak tides come from the same distribution as peak tides associated with extreme skew surges. Figure 2 highlights how the dependence between skew surge and peak tide is changing with the time of year; the distribution of peak tides and the peak tides associated with extreme skew surge are most different in May and least different in February. In May, the mode of the distribution of peaks tides associated with extreme skew surges has shifted to a lower value than the distribution of all peak tides. Results from the Anderson Darling test for every month tell us there is insufficient evidence to reject the null hypothesis that peak tides and the peak tides associated with extreme skew surge come from the same distribution in February, March, September, November and December. In these months, we conclude it is reasonable to assume skew surge and peak tide are independent. In the remaining months, we find sufficient evidence to suggest the two components are dependent. We believe this poses a really interesting area for further research.

In D'Arcy et al. (2021) we fit a GPD (see Coles (2001) for details) to extreme skew surges that accounts for seasonal variations, since they are more extreme in the winter. Our results show this is an improvement on a standard GPD fit, where skew surges are assumed to be independent of peak tide. Here, we investigate adding a covariate of peak tide into our skew surge model to account for the dependence found at Sheerness. D'Arcy et al. (2021) defines extreme skew surges as exceedances of the monthly 0.95 quantile. Non-stationarity is accounted for in the scale parameter of the GPD through a daily covariate $d = 1,...,365$:

$$\sigma_d = a + b\sin\left(\frac{2\pi}{365}(d - \phi)\right)$$

(3.1)

for $a,b,\varphi \in R$ parameters to be estimated. Note that the shape parameter of the GPD is fixed across months. To account for the dependence between skew surge and tide, we now also consider the following parameterisation on the scale parameter:

$$\sigma_{d,t} = a + b\sin\left(\frac{2\pi}{365}(d - \phi)\right) + ct$$

(3.2)

where $c \in R$ is another parameter to be estimated, and $t$ the associated peak tide observation. We fit a GPD with both (3.1) and (3.2) formulations to extreme skew surges at Heysham and Sheerness. The Akaike information criteria (AIC), frequently used for model selection, suggests that formulation (3.1) gives a better model fit at Heysham, i.e., an independence conclusion is supported by this analysis. Whereas, AIC suggests that the parametrisation in equation (3.2) yields a better fit at Sheerness. By ordering peak tide observations from smallest to largest and calculating $\sigma_{d,t}$ at its winter peak ($d = 365$) for each value, we observe a 15% reduction in the scale parameter as tides increase at Sheerness, which corresponds to an equal percentage reduction in the skew surge quantiles for excesses of the December skew surge threshold. We also compare the model fits at each site using a Likelihood Ratio Test. At Heysham the $p$ value is 0.657 which provides insufficient evidence to reject the null hypothesis, that is the simpler model (in equation (3.1)) is sufficient. Whereas, at Sheerness we get a much smaller $p$ value of 0.059, which provides statistically significant evidence at the 0.1 significance level to reject the null hypothesis and conclude that the more complex model is required. This highlights the importance of accounting for skew surge and peak tide dependence at sites where the independence assumption is not justified.

This is very interesting. Thank you for including it in your review. In the draft I have added a citation to your forthcoming publication.

**Ungauged Sites**

We feel you do not do full justice to your developments given the focus is on comparisons made at gauged sites with long and trustworthy records. The real value in modelled data is the ability to give estimates at other sites. This could be the focus of a natural follow up paper.

Have added a sentence in the "Suggestions for further work"

**Other Comments**

5. In the abstract, you say "results suggest an event of this magnitude has an expected frequency of about 1 in 500 years at [Sheerness]" when referring to the North Sea floods of 1953. In the paper, the only result to show this is presented in Section 6.2: "the fact that the 483-year surge-only simulation produces more than one event of comparable magnitude to the simulated 1953 event suggest the return period of the 1953 atmospheric forcing is less than 483 years." If this is the justification, it would be better to more formally quantify this using your statistical models.

   I can see several different ways to approach that. In the preceding sections we have advocated using only the shape parameter from the simulations. Using the observational location and scale parameters (as in Fig. 2 panels (a) and (b)) and the simulation-diagnosed shape parameter (panel (d)), the range of the Wadey skew surges as shown in our figure 7 correspond to return periods ranging between about 650 and 2000 years. (Note to self: code_ABR). Having said that, I feel that this level of detail is not appropriate in this preliminary "Towards using state-of-the-art climate models…" type of publication, so I have cut the reference to the expected frequency out of the abstract, and cut the corresponding short paragraph from the main text.

6. In the Introduction (line 51) you list assumptions required to fit extreme value models as 'events are effectively random and statistically independent of each other.' But what about events being identically distributed (or stationary), it is clear that tide and storm surge (or skew surge) are not stationary as both process exhibit seasonality. Instead of "effectively random" it would be more mathematically correct to say "stochastic." Extreme value methods also handle dependence, so "independent of each other" is not formally required.

   Rephrased as follows:

   The statistical models which are fitted to the observational data in order to infer the levels of unprecedented extremes are supported by mathematical arguments which may require assumptions such as the assumption that the events are stochastic. We know that the real-world events are deterministic, and furthermore may be auto-correlated over a range of timescales. Such auto-correlation can be accounted for within the statistical framework, for example by the use of an extremal index~\citep{Tawn1992estimating, Batstone2013UK}. Alternatively, a physically-based numerical model has the potential to directly address both determinism and auto-correlation by simulating them.

7. On line 84 climate change is discussed. Tide gauge observations will exhibit an approximately linear mean trend due to sea level rise, it is unclear whether this has been removed before comparison with the simulations in Section 4. It is likely that removing this change will not change the results significantly, but it is important to remove this non-stationary effect before fitting extreme value models.

   Have added this phrase:

   "The trend due to sea level rise can be seen in the tide gauge observations and was carefully removed before making a statistical fit to the extremes. For details see CFB2018. Our numerical model of the shelf sea does not include any change in mean sea level."

8. Figure C1 shows the reduction in uncertainty on shape parameter when record lengths are increased in the tide gauge network, but it would be nice to see this for more record lengths that are equally spaced

(say 10 to 500 years in increments of 10 years) to really highlight this - with measures of uncertainty as in Figure C1 (d).

Good idea. Done. Thanks.

**Technical Corrections**

10. Table 1 gives a list of useful acronyms and symbols, it would help to include what 'GC3' and 'MM' stand for in 'HadGEM3-GC3-MM' as it is not mentioned in text.

Thank you for pointing this out. Have added it to Table 1.

11. Equation (2) has a surplus close bracket. Corrected, thanks.

12. Equation (6), should this derivative be evaluated at $L = \log(u)$ rather than $y = u$ since $y$ is the return level (as in equation (1))?

No, I don't think so. However I acknowledge that, in equation 6, the point on the RL curve at which to evaluate the gradient is specified in a non-standard way: in terms of the ordinate ($y$) instead of the usual specification in terms of the abscissa ($L$). Your suggested phrase "evaluated at $L = \log(u)$" does not mean the same thing. Please see the uploaded supplement **equation6_revisited.pdf** for further explanation.

**Note to self: upload eq 6 supplement.**

13. Figures 2 (c) and (d) would benefit from having 95% confidence intervals on shape parameters using the model data.

Have added these, and commented in the text on the added value in terms of certainty that the CFB method provides compared to a simple GEV fit to annual maxima --- this is well-illustrated by the confidence intervals shown.

14. Line 283: We think the wrong figure is referenced here.

Please note this is figure E.1 in the CFB technical publication ("**their** figure E.1", as stated in the text). In the copy I downloaded from here:

https://assets.publishing.service.gov.uk/media/603652cce90e0740b7caac9d/Coastal_flood_boundary_conditions_for_the_UK_2018_update_-_technical_report.pdf

figure E.1 is on page 64, titled "Estimated shape parameter for the UK tide gauges."

15. Line 319: "They used ....time-series." This has nothing to do with how CFB2018 estimate the shape parameters and should be cut. It only links to the derivation of SWL return levels.

Have cut these two sentences as advised.

16. Line 349: You hint that the reason for the disagreement between the methods could be due the short observed data series. But Figure B1 shows major departures between the observed data and the model data in the body of the distributions to a level that could in no way be due to short observed series and must be that the model data does not reproduce well the observed data. Linked to this, in discussing Figure 1 you say the agreement at both sites is "excellent". With the amount of data at Sheerness it is clear that there are major disagreements that cannot be accounted by sampling variations.

Have reworded that paragraph to be less bullish:

"Figure 1 shows good model vs observations agreement at Workington, and even these two sites alone illustrate that our modelling system is able to simulate unprecedented skew surge events (i.e. events of a magnitude not found in the tide-gauge record). However, the quality of agreement shown at Workington

is not exhibited everywhere. Empirical return level plots of skew surge for a set of 44 tide gauge locations around the UK coastline are shown in the appendix in Fig. B1. This gives a qualitative, visual sense of the realism of the model in terms of the simulated extremes. The good agreement at Workington can be contrasted with the poor agreement at, for example, Newlyn or Aberdeen, where the simulated extremes are negatively biased relative to the corresponding observations. The model does not give higher quantiles than the observed data at any site. We argue (below) that the simulation may nevertheless be able to add value to estimations of unprecedented events, even where a bias exists."

17. Line 421: You suddenly mention a "kernel" to spread the duration of events but fail to provide any information. A little detail would be helpful here.

I have added the following sentences explaining the choice and purpose of the kernel:
"The kernel was designed to represent the important features of the RACMO-driven surge-only simulation, i.e., the approximate duration and shape of the time series plot. The purpose of convolution with the kernel is to identify those events which correlate well (in terms of their time series plot) with the RACMO-driven simulation, in other words, events which not only produce a large surge, but are also of comparable duration to the RACMO-driven simulation. The kernel was not used to modify events, but simply to identify significant ones."

18. Figure E2: A description of the different points would be useful.

Have added the following text:  "The blue points show simulated skew surge against simulated astronomical high water for the 16 extreme atmospheric events with timing adjusted such that the event coincided with a simulated spring tide and a simulated neap tide, as described above. The artificial case of no tide is also shown. Grey points are as in Williams (2016)."


AC4:

This was an earlier draft of AC5, including a misleading typo. Have included AC5 instead here.

AC5:

**Equation 6 revisited**

**Tom Howard**

**September 9, 2021**

**Intro**

The purpose of this document is to give a more detailed explanation of Equation 6 of "Towards using state-of-the-art climate models to help constrain estimates of unprecedented UK storm surges", Howard and Williams (2021).

**Statistical Modelling of Extreme Values**

To identify, for example, the 1000-year return level based solely on tide-gauge observations, some philosophy for making out-of-sample estimates is required. The usual approach is to exploit the most extreme observations, and theories concerning their behaviour, under some restrictive assumptions.

**Annual Maxima**

One popular and simple approach is fitting a Generalised Extreme Value (GEV) distribution to the annual maxima. The GEV distribution (GEVD) arises as the limiting case for block maxima as the block size tends to infinity. In the case of annual maxima, "block" means one year. The GEVD is characterised by three parameters. For readers unfamiliar with the GEVD, it may be helpful to picture the effect of these parameters in terms of a return-level curve, such as the ones shown in Fig. **??**. The location parameter, $\mu$, is comparable to an intercept. An increase in $\mu$ slides the whole curve up the Y-axis. $\mu$ is the Y-value (return level) evaluated at the one-year return period:

$$\mu = y\Big|_{L=0}$$

where $L = \log(\text{return period})$ and $y$ is the return level. Notice that, though not particularly useful, this could be written

$$\mu = y\Big|_{y=\mu}$$

The GEV scale parameter, $\sigma$, is the gradient of the curve, evaluated at the one-year return period. This could either be written as

$$\sigma = \frac{dy}{dL}\Big|_{L=0} \tag{1}$$

or, for comparison with equation 6,

$$\sigma = \frac{dy}{dL}\Big|_{y=\mu}$$

Since $y$ is a monotonic function of $L$, and in view of the first (unnumbered) equation, this is an alternative way to unambiguously define the point on the RL curve at which to evaluate the gradient. It's just specified in a non-standard way: in terms of the ordinate ($y$) instead of the usual specification in terms of the abscissa ($L$).

The shape parameter, $\xi$, determines the curvature. Negative $\xi$ corresponds to a curve which flattens out at high return periods, approaching an upper bound as the return period tends to infinity. With positive $\xi$ the curve has no upper bound, but has a lower bound as the return level decreases. When $\xi = 0$ the curve is a straight line and has neither lower nor upper bound. This follows the convention of [**?**] for the shape parameter. However, not all sources follow this convention. In CFB2018, "shape parameter" refers to the negative of our $\xi$. In the wider literature the "shape parameter" may refer to the negative or the reciprocal of our $\xi$. To make our shape parameter notation unambiguous: if $Y$ is a random variable with GEV distribution, our shape parameter $\xi$ is defined such that the distribution of $Y$ is given by

$$P(Y < y) = \exp\left\{-\left[1 + \xi\left(\frac{y - \mu}{\sigma}\right)\right]^{-1/\xi}\right\} \tag{2}$$

This can be more simply expressed as the corresponding return level curve, which is

$$\frac{y - \mu}{\sigma} = \frac{R^\xi - 1}{\xi} \tag{3}$$

where the average recurrence interval (or "return period") is $R$ and the corresponding return level is $y$. The connection between equations 2 and 3 is seen by regarding exceedances of the $R$-year return level $y$ as Poisson-distributed random occurrences, occurring at an average rate

$$\lambda = 1/R \qquad (4)$$

The probability of no such occurrences in a given year is then given by standard Poisson statistics:

$$P(\text{no occurrences}) = P(Y < y) = \exp(-\lambda) \qquad (5)$$

Combining 3, 4 and 5 gives equation 2. The particular case $\xi = 0$ is obtained by taking the limit as $\xi \to 0$.

**Peaks over Threshold**

The most extreme storm surges in the UK are caused by the storminess of the winter atmosphere, so the annual maximum event is always expected to occur in winter. Thus, an advantage of the annual-maxima approach described above is that the annual maxima are typically very well separated from each other and thus can be considered independent, particularly if the nominal year change is taken to be in the summer. A disadvantage of the approach is that it uses only the annual maxima. On the other hand, the peaks-over-threshold (POT) approach uses all of the data exceeding a chosen threshold. This formed part of the approach taken by CFB2018. An advantage of this approach is that, if a low-enough threshold is used, it has the potential to exploit more of the available data (i.e. an average of more than one extreme event per year), whilst including only extreme events. Such exploitation of more data usually reduces the uncertainties in inferred statistics (e.g. the out-of-sample estimates). This is particularly desirable when short observational records limit the available extremes. However, if the threshold is too low, some of the data included can no longer be considered "extreme" and may bias the result. This is the wellrecognised bias-variance trade-off. Another disadvantage is that including more than one event from a winter may compromise the independence of the events. (Skew surge can be evaluated for every high tide, and a weather system can generate a substantial skew surge on successive high tides.) Dependence is accommodated by CFB2018 using an extremal index... For a detailed comparison of the annual-maxima and POT approaches see...

The usual POT approach is to fit a Generalised Pareto Distribution (GPD) to the peaks. The GPD has two parameters. The shape parameter $\xi$ is shared with the GEVD. The GPD scale parameter, $\sigma$, is the gradient of the plot of e
return level against log of return period at the return period of the chosen threshold, $u$,

$$\widetilde{\sigma} = \frac{dy}{dL}\bigg|_{y=u} \qquad (6)$$

As in the unnumbered equation following equation 1, the point on the RL curve at which to evaluate the gradient is specified in a non-standard way: in terms of the ordinate ($y$) instead of the usual specification in terms of the abscissa ($L$). $\sigma$ is a property of both the extreme value distribution and the chosen threshe old. The GEV scale parameter, $\sigma$, on the other hand, is a property of the extreme value distribution only and is thus a more fundamental parameter for making comparisons: it can be used in a like-for-like comparison of the results of different thresholds, or for comparison of GEV and GPD results. The two different scale parameters are related by $\sigma = \widetilde{\sigma}\lambda_u^{\xi}$, where $\lambda_u$ is the expected number of exceedances of $u$ per year.

Though not formally a parameter of the GPD, a threshold must be chosen. CFB2018 tested 14 different thresholds and, finding no clear support for dismissal of any, elected to evaluate statistics based on each threshold and identify the median as the best estimate.
* * *
RC4:

Thank you for the positive way you have responded to our suggestions and also for the extra clarification on the return level curve connections with the scale parameter.

Eleanor D'Arcy and Jonathan Tawn
* * *
AC5:   Thank you again for your review.

Latexdiff showing differences between original submission and revised submission following all review comments follows (assuming I manage to combine it successfully)…

[revised manuscript text omitted]

**4.1 Quantitative evaluation of simulation of extremes**

To make some quantification of the realism of the simulated extremes, we used the statistical models described in §3.3 to fit the simulated extremes. We fitted a GEV model (§3.3) to the simulated annual maxima using the MLE method (§3.3). We fitted the model pointwise (that is, for each tide gauge we fitted independently at a model grid cell closest to the tide-gauge of that site; this model grid cell is taken to represent that site). This gives a spatial distribution of diagnosed parameters. We find  good agreement between the simulation-based and tide-gauge-based location and scale parameters (Fig. 2). We also find a surprisingly good correlation between the spatial distribution of simulation-diagnosed shape parameters and the corresponding spatial distribution of shape parameters diagnosed by CFB2018. Pearson's r for the shape parameter correlation is 0.72 when we use a GEV fit to the simulated annual maxima, and 0.86 when we use a GPD fit to the simulated peaks over a threshold (see §4.2).

[Figure]

**Figure 2.** Comparison of simulation-based and observation-based skew surge extreme value distribution parameters. (a): Location parameter. (b) GEV scale parameter ($\sigma$). (c, d): Shape parameter. The correlation seen in all panels shows that the model successfully simulates the observed spatial variations in the extremes. Pointwise uncertainties in the estimated shape parameters are included in panels (c) and (d). The CFB uncertainty shown is the 95% confidence interval of the GPD fit at a 95% threshold. In panel (d) (the last panel), the simulation uncertainty shown is evaluated in the same way. In panel (c) (next-to-last), the simulation uncertainty shown is the 95% confidence interval of the GEV fit to the simulated annual maxima. For  further details see main text.

[revised manuscript text omitted]

355 Given the need for some kind of constraint on the shape parameter when fitting observational records, use of shape parameters from a long simulation holds the promise of reducing uncertainties. For example, if we assume that the model-diagnosed spatial pattern of shape parameters is correct but uniformly biased by a scalar $\xi_{bias}$ (which does not vary over sites), $\xi_{true}(x) = \xi_{model}(x) + \xi_{bias}$ where $x$ is a vector of sites, then we can use the observations from *all sites* to estimate the one scalar parameter $\xi_{bias}$. This could lead to substantial reductions in the uncertainty 360 of $\xi_{true}$ estimates over $x$.

[revised manuscript text omitted]

**8   Suggestions for Further Work**

The model/observation departures seen in Fig. B1 have a systematic feature which is consistent over spatial regions, e.g., south-west and north-west UK. This suggests that it should be possible to account for these departures through a smooth spatial function which maps the differences in quantiles between the observations and model data. With this adjustment it is possible that the currently identified under-estimation may be corrected before making the tail-based GEV/GPD comparisons shown here.

The shape parameter estimates in Fig. 2(d) show some site-to-site variations which are more pronounced than the broader smooth variations across coastlines. This suggests that they would also benefit from the penalty-based approach used by CFB2018.

One advantage of modelled data is the ability to give estimates at ungauged sites. We have not exploited this ability here, but we anticipate that it will form the basis of further work.

*Data availability.* The tide gauge data used in the CFB2018 report are available to download from the National Tidal and Sea Level Facility (ntslf.org). The CFB2018 shape parameters can be read from their figure E.1 (Environment Agency, 2018). The CFB2018 GEV scale parameters as shown in Fig. 2(b), in metres, (for sites Newlyn... Portrush in the order shown on the X-axis of Fig. 2) are:

0.0835, 0.0733, 0.0847, 0.1031, 0.1369, 0.1790, 0.1574, 0.1484, 0.1140, 0.0940, 0.1639, 0.1063, 0.1161, 0.1305, 0.1228, 0.1815, 0.1593, 0.1358, 0.1757, 0.1490, 0.1471, 0.1202, 0.0955, 0.1368, 0.0684, 0.0913, 0.0922, 0.0966, 0.1049, 0.1178, 0.1333, 0.1618, 0.1748, 0.2153, 0.1715, 0.1629, 0.1557, 0.1091, 0.0985, 0.0937, 0.0948, 0.1201, 0.0938, 0.1273

Simulated sea levels at the tide gauge sites as used in our analysis are available from the first author on request. All of the analysis was undertaken using the open source languages R and Python.

**Appendix A: Surge model grid and tide gauge locations**

[Figure]

**Figure A1.** (a) Domain and grid of the CS3 coastal shelf model. Grid size is 1/9 degree in latitude and 1/6 degree in longitude, which results in near-square grid cells at the latitude of the UK. (b) Tide gauge locations.

**Appendix B: Empirical return level plots for UK tide gauges**

Figure B1 shows empirical return level plots for 44 tide gauge locations around the UK. For simplicity we use annual maxima only. The observational annual maxima are limited to years in which the tide gauge data is at least 75 % complete. Plotting positions are evaluated using the Weibull formula (Weibull, 1939). Working from left to right along the rows (and then downwards through the rows as in reading), the sequence of plots follows the clockwise sequence of Fig. 2 as described in §4.1.

[Figure]

**Figure B1.** Empirical return level plots for 44 UK tide gauges.  Purple shows observational (tide-gauge) data.  Blue shows data from the 483-year HadGEM3-GC3-MM simulation.

 Figure B1 shows some major departures between the model and observed data across the distribution of skew surges, but particularly in the tails. The model does not give higher quantiles than the observed data at any site. However, it can be seen
530  that at some locations the model produces a plausible simulation of the observed return level plot and a plausible extrapolation of the return level plot to return periods outside of the observational record. This is discussed further in §4.

**Appendix C: Shape-parameter uncertainty dominance**

For short record lengths, unconstrained maximum-likelihood estimation is known to give "noisy" and implausible shape parameters (Coles and Dixon, 1999; Martins and Stedinger, 2000), see also §3.3. We illustrate this in Fig. C1 with a GEV fit to
535  tide-gauge data.

[Figure]

**Figure C1.** Short record lengths lead to noisy MLE shape-parameter estimates. (a) Shape parameter of GEV fit to tide gauge data against the number of annual maxima fitted (blue dots, one for each tide gauge), and shape parameter of GEV fit to model data (orange crosses, one for each tide gauge, all 483 years). Note that the range of fitted shape parameters reduces ("tapers") as record length increases. (b) Model data for each port is cut down to a (random) sub-sample having the same length as the observational record at that port and then fitted in the same way as the observations. A similar tapering pattern emerges. (c) as (b) but a different random sub-sample.  Also shown (grey) is the 5 to 95 percentile range from one hundred such sub-samples at each port. (d) Similar to (a, b, c) but using pseudo-random variates drawn from a GEV distribution with scale and shape parameter which are typical of the values found around the UK (0.12 metres and -0.06 respectively). Sample size varies from 5 to 500, as shown by the logarithmically-scaled x-axis. For each sample, GEV parameters are fitted by maximum likelihood estimation. For each sample size, 2000 samples are drawn and the 5 to 95 percentile of the fitted shape parameters is shown by the grey bar. The dashed green line shows a shape of -0.06. The 5 to 95 percentile intervals for sample sizes 5, 7, and 10 exceed the Y-axis limits.

[revised manuscript text omitted]
. The blue points show simulated skew surge against simulated astronomical high water for the 16 extreme atmospheric events with timing adjusted such that the event coincided with a simulated spring tide and a simulated neap tide, as described above. The artificial case of no tide is also shown. Grey points are as in Williams et al. (2016). It can be seen that our model results do not look out of context compared to

600    the observations in terms of the negative correlation. For reference the extreme (entirely artificial) case of simulated tide-surge interaction is also shown, in which no tidal forcing is included (see §6.1). In reality, of course, the tide is always present.

[Figure]

**Figure E2.** Our skew-surge/tide interaction results for Sheerness overlain on a reproduction of Williams et al.(2016), their figure S2. The results for a modelled high water of zero are included for comparison. They come from a very artificial simulation in which no tidal forcing is included. In reality, of course, the tide is always present.

*Author contributions.* TH devised the experiment, peformed most of the new analysis, and wrote the article. SDW performed the original CFB2018 analysis on the tide gauge data and replicated it for the simulation.

*Competing interests.* The authors declare no competing interests.

605 *Acknowledgements.* This work was supported by the Met Office Hadley Centre Climate Programme funded by BEIS and Defra. We thank Andreas Sterl, Eleanor D'Arcy, and Jon Tawn for their generous reviews, which helped to improve the initial submitted manuscript of this work. Special thanks to Jenny Sansom for helpful discussions and for providing some of the data used in the evaluation. Thanks to Erik van Meijgaard  at KNMI for comments on an early draft and for permission to use their RACMO-based atmospheric reconstruction of the 1953 event, and to Mark Pickering at National Oceanography Centre Southampton for helping to supply the data.

610 Thanks to Graham Siggers (HR Wallingford) and Matt Palmer (Met Office Hadley Centre) for helpful comments on an early draft of this paper. Thanks to Jeff Ridley for help with the HadGEM3-GC3-MM data. Thank you to  Simon Brown and Rob Shooter for helpful discussions and guidance with extreme value modelling. This publication contains public sector information licensed under the Open Government Licence v3.0.